# What should we do when a model crashes? Recommendations for global sensitivity analysis of earth and environmental systems models

Razi Sheikholeslami[1,2], Saman Razavi[1,2,3], Amin Haghnegahdar[1,2]

[1]School of Environment and Sustainability, University of Saskatchewan, Saskatoon, Canada
[2]Global Institute for Water Security, University of Saskatchewan, Saskatoon, Canada
[3]Department of Civil, Geological, and Environmental Engineering, University of Saskatchewan, Saskatoon, Canada

*Correspondence to*: Razi Sheikholeslami (razi.sheikholeslami@usask.ca)

**Abstract.** Complex, software intensive, technically advanced, and computationally demanding models, presumably with ever-growing realism and fidelity, have been widely used to simulate and predict the dynamics of the Earth and environmental systems. The parameter-induced simulation crash (failure) problem is typical across most of these models despite considerable efforts that modellers have directed at model development and implementation over the last few decades. A simulation failure mainly occurs due to the violation of the numerical stability conditions, non-robust numerical implementations, or errors in programming. However, the existing sampling-based analysis techniques such as global sensitivity analysis (GSA) methods, which require running these models under many configurations of parameter values, are ill equipped to effectively deal with model failures. To tackle this problem, we propose a new approach that allows users to cope with failed designs (samples) when performing GSA without re-running the entire experiment. This approach deems model crashes as missing data and uses strategies such as median substitution, single nearest neighbour, or response surface modelling to fill in for model crashes. We test the proposed approach on a 10-parameter HBV-SASK rainfall-runoff model and a 111-parameter MESH land surface-hydrology model. Our results show that response surface modelling is a superior strategy, out of the data filling strategies tested, and can comply with the dimensionality of the model, sample size, and the ratio of the number of failures to the sample size. Further, we conduct a "failure analysis" and discuss some possible causes of the MESH model failure that can be used for future model improvement.

## 1 Introduction

### 1.1 Background and motivation

Since the start of the digital revolution and subsequent increase in computers' processing power, the advancement of information technology has led to significant development of the modern software programs for Dynamical Earth System Models (DESMs). The current-generation DESMs typically span upwards of several thousand lines of code and require huge amounts of data and computer memory. The flip side of the growing complexity of the DESMs is that running these models will pose many types of software development and implementation issues such as simulation crashes/failures. The simulation

crash problem happens mainly due to violation of the numerical stability conditions needed in DESMs. Certain combinations of model parameter values, improper integration time step, inconsistent grid resolution, or lack of iterative convergence as well as model thresholds and sharp discontinuities in model response surfaces, all associated with imperfect parameterizations, can cause numerical artefacts and stop DESMs from properly functioning.

When model crashes occur, the accomplishment of automated sampling-based model analyses such as sensitivity analysis, uncertainty analysis, and optimization becomes challenging. These analyses are often carried out by running DESMs for a large number of parameter configurations randomly sampled from a domain (parameter space) (see, e.g., Raj et al., 2018; Williamson et al., 2017; Metzger et al., 2016; Safa et al., 2015). In such situations, for example, the model's solver may break down because of the implausible combinations of parameters ("unlucky parameter set" as termed by Kavetski et al., (2006)),

failing to complete the simulation. It is also possible that a model will be stable against perturbation of a single parameter, while it may crash when several parameters are perturbed simultaneously. "Failure analysis" is a process that is performed to determine the causes that have led to such crashes while running DESMs. Before achieving a conclusion on the most important causes of crashes, it is necessary to check the software code of the DESMs and confirm if it is error-free (e.g., if a proper numerical scheme has been adopted and correctly coded in the software). This often requires investigating both the software

documentation and a series of nested modules. However, the existence of numerous nested programming modules in typical DESMs can make the identification and removal of all software defects so tedious. In addition, as argued by Clark and Kavetski (2010), the numerical solution schemes implemented in DESMs are sometimes not presented in detail. This is one important reason why detecting the causes of simulation crashes in DESMs is usually troublesome. For example, Singh and Frevert (2002) and Burnash (1995) described the governing equations of their models without explaining the numerical solvers that

were implemented in their codes.

       Importantly, the impact of simulation crashes on the validity of global sensitivity analysis (GSA) results has often been overlooked in the literature, where simulation crashes have been commonly classified as ignorable (see section 1.2). As such, a surprisingly limited number of studies have reported simulation crashes (examples related to uncertainty analysis include Annan et al., 2005; Edwards and Marsh, 2005; Lucas et al., 2013). This is despite the fact that these crashes can be very

computationally costly for the GSA algorithms because they can waste the rest of the model runs, prevent completion of GSA, or inevitably introduce ambiguity into the inferences drawn from GSA. For example, Kavetski and Clark (2010) demonstrated how numerical artefacts could contaminate the assessment of parameter sensitivities. Therefore, it is important to devise solutions that minimize the effect of crashes on GSA. In the next subsection, we critically review the very few strategies for handling simulation crashes that have been proposed in the literature and identify their shortcomings.

**1.2 Existing approaches to handling simulation crashes in DESMs**

We have identified, as outlined below, four types of approaches in the modelling community to handle simulation crashes. The first two are perhaps the most common approaches (based on our personal communications with several modellers); however, we could not identify any publication that formally reports their application:

1. After the occurrence of a crash, modellers commonly adopt a conservative strategy to address this problem by altering/reducing the feasible ranges of parameters and re-starting the experiment in a hope to prevent recurrence of the crashes in the new analyses.

2. Instead of GSA that runs many configurations of parameter values, analysts may choose to employ local methods such as local sensitivity analysis (LSA) through running the model only near the known plausible parameter configurations.

3. Some modellers may adopt an ignorance-based approach by using only a set of ''good'' (or behavioural) outcomes/responses in sampling-based analyses and ignoring unreasonable (or non-behavioural) outcomes such as simulation crashes. This can be done in conjunction with defining a performance metric to choose which simulations to exclude from the analysis (see, e.g., Pappenberger et al., 2008; Kelleher et al., 2013).

4. The most rigorous approach seems to be a non-substitution approach that tries to predict whether or not a set of parameter values will lead to a simulation crash. Webster et al. (2004), Edwards et al. (2011), Lucas et al. (2013), Paja et al. (2016), and Treglown (2018) are among few studies that aimed at developing statistical methods to predict if a given combination of parameters can cause a failure. For example, Lucas et al. (2013) adopted a machine learning method to estimate the probability of crash occurrence as a function of model parameters. They further applied this approach to investigate the impact of various model parameters on simulation failures. A similar approach is based on model pre-emption strategies, where the simulation performance is monitored while the model is running and the model run is terminated early if it is predicted that the simulation will not be informative (Razavi et al. 2010; Asadzadeh et al., 2014).

The above approaches have some major limitations in handling simulation crashes in the GSA context, because:

1. Locating regions of the parameter space responsible for crashes (i.e., "implausible regions") is difficult and requires analysing the behaviour of the DESMs throughout the often high-dimensional parameter space. Implausible regions usually have irregular, discontinuous, and complex shapes, and thus are too effortful to identify. Additionally, altering/reducing the parameter space by excluding the implausible regions changes the original problem at hand.

2. It is well known that local methods (e.g., LSA) can provide inadequate assessments that can often be misleading (see e.g., Saltelli and Annoni, 2010, Razavi and Gupta, 2015).

3. Ignoring the crashed runs in GSA may only be seen relevant when using purely random (and independent) samples (i.e., Monte Carlo method). In such cases, if the model crashes at a given parameter set, one may simply exclude that parameter set or generate another random parameter set (at the expense of increased computational cost) that results in a successful simulation.

4. Some efficient sampling techniques follow specific spatial arrangements; examples include the variance-based GSA proposed by Saltelli et al. (2010) or STAR-VARS of Razavi and Gupta (2016b). In GSA enabled with such structured sampling techniques, we cannot ignore crashed simulations because excluding sample points associated with simulation

crashes will distort the structure of the sample set, causing inaccurate estimation of sensitivity indices. As a result, the user may have to re-do a part or the entire experiment depending on the GSA implementation.

5. Implementation of the non-substitution procedures necessitates significant prior efforts to identify a number of model crashes based on which a statistical model can be built, so as to predict and avoid simulation failures in the subsequent model runs. Such procedures can easily become infeasible in high-dimensional models, as they would require an extremely large sample size to ensure an adequate coverage of the parameter space for characterizing implausible regions and building a reliable statistical model. These strategies can be more challenging when a model is computationally intensive. For example, to determine which parameters or combinations of parameters in a 16-dimensional climate model were predictors of failure, Edwards et al. (2011) used 1,000 evaluations (training samples) for constructing a statistical model to identify parameter configurations with high probability of failure in the next 1,087 evaluations (2,087 model runs in total). As pointed out by Edwards et al. (2011), although 2,087 evaluations might impose high computational burdens, a much larger sample size spreading out over the parameter space is required to guarantee reasonable exploration of the 16-dimensional space.

These shortcomings and gaps motivated our investigation to develop effective and efficient crash handling strategies suitable for GSA of the DESMs, as introduced in section 2.

**1.3 Scope and outline**

The primary goal of this study is to identify and test practical "substitution" strategies to handle the parameter-induced crash problem in GSA of the DESMs. Here, we treat model crashes as missing data and investigate the effectiveness of three efficient strategies to replace them using available information rather than discarding them. Our approach allows the user to cope with failed simulations in GSA without knowing where they will take place and without re-running the entire experiment. The overall procedure can be used in conjunction with any GSA technique. In this paper, we assess the performance of the proposed substitution approach on two hydrological models, by coupling it with a variogram-based GSA technique (VARS; Razavi and Gupta (2016a,b)).

The rest of the paper is structured as follows. We begin in the next section by introducing our proposed solution methodology for dealing with simulation crashes. In section 3, two real-world hydrological modelling case studies are presented. Next, in section 4, we evaluate the performance of the proposed methods across these real-world problems. The discussion is presented in section 5, before drawing conclusions and summarizing major findings in section 6.

## 2 Methodology

### 2.1 Problem statement

We denote the output of each model run (realization) $y(X)$, which corresponds to a $d$-dimensional input vector $X = \{x_1, x_2, \dots, x_d\}$, where $x_i$ $(i = 1, 2, \dots, d)$ is a factor that may be perturbed for the purpose of GSA (e.g., model parameters,

initial conditions, or boundary conditions). Running a GSA algorithm usually requires generating $n$ realizations of a simulation model using an experimental design $X^s = \{X_1, X_2, \dots, X_n\}^T$ forming a $n \times d$ sample matrix. Then, the model responses will form an output space as $Y = \{y(X_1), y(X_2), \dots, y(X_n)\}^T$. Here, we deem simulation crashes as missing data and consider the model mapping of $X^s \rightarrow Y$ as an incomplete data matrix. For a given $Y \in \Re^{1 \times n}$ with missing values, let the vector $Y_a$ consist of the $n_a$ locations in the input space for which, in the given $Y$, the model responses are available, and let the vector $Y_m$ consist

of the remaining $n_m$ locations ($n_m = n - n_a$) for which, in the given $Y$, the model responses are missing due to simulation crashes. For convenience of expression and computation, we use the "$NAN_j$" symbol to represent the $j$th missing value in vector $Y$. The main goal now is to develop and test data recovery methods that can be used to substitute model crashes $Y_m$ using available information (i.e., $Y_a$ and $X^s$).

### 2.2 Proposed strategy for handling model crashes in GSA

We propose and test three techniques adopted from the "incomplete data analysis" for missing data replacement– the process known as imputation (Little and Rubin, 1987). Our techniques do not account for the mechanisms leading to crashes because identifying such mechanisms can be very challenging (Liu and Gopalakrishnan, 2017). Therefore, only the non-missing responses and the associated sample points are included in our analysis to infill model crashes for GSA, as described in the next sub-sections.

### 2.2.1 Median substitution

In sampling-based optimization, one may assign a very poor objective function value (e.g., a very large objective function in the minimization case) to a crashed solution, similar to the big M method for handling optimization constraints (Camm et al., 1990). Our first strategy in the GSA context adopts such an approach. However, since replacing crashes with a big value can magnify the effect of the crashed runs in GSA, instead we suggest choosing a measure of central tendency such as mean or

median to minimize the impact of the implausible parameter configurations on the GSA results. If the distribution of the model responses is not highly skewed, imputing the crashes with the mean of the non-missing values may work. However, if the distribution exhibits skewness, then the median may be a better replacement because the mean is sensitive to the outliers. Therefore, we use the median substitution technique for the experiments reported in this paper. In general, this strategy treats each model response as a realization of a random function and ignores the covariance structure of the model responses. Also,

a shortcoming of this technique is that while it preserves the used measure of central tendency of $Y$, it can distort other statistical properties of $Y$, for example by reducing its variance.

## 2.2.2 Nearest neighbour substitution

The Nearest Neighbour (NN) technique (also known as hot deck imputation, see, e.g., Beretta and Santaniello (2016)) uses observations in the neighbourhood to fill in missing data. Let $X_j \in X^s$ be an input vector for which a simulation model fails to return an outcome. Basically, in the NN-based techniques, $NAN_j$ is replaced by either a response value corresponding to a single nearest neighbour (single NN) or a weighted average of the response variables corresponding to $k$ nearest neighbours ($k$-NN) where $k > 1$. The underlying rationale behind the NN-based techniques is that the sample points closer to $X_j$ may provide better information for imputing $NAN_j$. In the $k$-NN techniques, weights are assigned based on the degree of similarity between $X_j$ and the $k$th nearest neighbour $X_k$, where $y(X_k) \in Y_a$, characterized through kernel functions (Tutz and Ramazan, 2015).

In this study, we choose to use the single NN technique with Euclidean distance measure. We do so because the single NN technique is very parsimonious and simple to understand and implement. To substitute the crashed simulations, the single-NN algorithm reads through whole dataset to find the nearest neighbour and then imputes the missing value with the model response of that nearest neighbour. It is noteworthy that some authors have asserted that covariances among $Y$ variables are preserved in the NN-based techniques when using small $k$ values (Hudak et al., 2008; McRoberts et al., 2002; Tomppo et al., 2002). But, McRoberts (2009) showed that the variance and covariance of the $Y$ variables tend to be preserved for $k = 1$ but not for $k > 1$ (McRoberts, 2009). In general, compared to the single NN-technique, the $k$-NN technique may provide a better fit to data but at the expense of being more complex and requiring a careful (and subjective) selection of the kernel functions and variable $k$. As a more complex technique, we suggest directly using a model emulation technique as described in the section below.

## 2.2.3 Model emulation-based substitution

Model emulation is a strategy that develops statistical, cheap-to-run surrogates of response surfaces of complex, often computationally intensive models (Razavi et al., 2012a). Here we develop an emulator $\hat{y}(.)$, which is a statistical approximation of the simulation model based on a response surface modelling concept. This strategy consists in finding an approximate/surrogate model with low computational cost that fits the non-missing response values $Y_a$ to predict the fill-in values for the missing responses $Y_m$. There are various types of response surface surrogates, which have been extensively discussed in the literature (see, e.g., Razavi et al., 2012a). Examples are polynomial regression, radial basis functions (RBF), neural networks, kriging, support vector machines, and regression splines. Here, we employ the RBF approximation as a well-established surrogate model. It has been shown that RBF can provide an accurate emulation for high-dimensional problems (Jin et al., 2001; Herrera et al., 2011), particularly when the computational budget is limited (Razavi et al., 2012b). An RBF model as a weighted summation of $n_a$ basis functions (and a polynomial or constant value) can approximate the predictive response $\hat{y}(X)$ at a sample point $X$ as follows:

$$\hat{y}(\boldsymbol{X}) = \sum_{i=1}^{n_a} \omega_i f(\|\boldsymbol{X} - \boldsymbol{X}_i\|) = \boldsymbol{f}(\boldsymbol{X})\boldsymbol{\omega} \tag{1}$$

where $\boldsymbol{f} = \{f_1, f_2, \ldots, f_{n_a}\}$ is the vector of the basis functions, $\omega_i$ is the $i$th component of the radial basis coefficient vector $\boldsymbol{\omega} = \{\omega_1, \omega_2, \ldots, \omega_{n_a}\}^{\mathrm{T}}$, and $\|\boldsymbol{X} - \boldsymbol{X}_i\|$ is the Euclidian distance between two sample points.

There are various choices for the basis function, such as Gaussian, thin-plate spline, multi-quadric, and inverse multi-quadric
5 (Jones, 2001). In the present study, we utilize the well-known Gaussian kernel function for RBF:

$$f(\|\boldsymbol{X} - \boldsymbol{X}_i\|) = exp\left(\frac{\|\boldsymbol{X} - \boldsymbol{X}_i\|^2}{c_i^2}\right) \tag{2}$$

where $c_i$ is the shape parameter which determines the spread of the $i$th kernel function $f_i$.

After choosing the form of the basis function, the coefficient vector $\boldsymbol{\omega}$ can be obtained by enforcing the accurate interpolation condition, i.e.:

$$10 \quad \begin{bmatrix} y(\boldsymbol{X}_1) \\ y(\boldsymbol{X}_1) \\ \vdots \\ y(\boldsymbol{X}_{n_a}) \end{bmatrix} = \begin{bmatrix} f_{11} & f_{12} & \cdots & f_{1n_a} \\ f_{21} & f_{22} & \cdots & f_{2n_a} \\ \vdots & \vdots & \ddots & \vdots \\ f_{n_a1} & f_{n_a2} & \cdots & f_{n_an_a} \end{bmatrix} \begin{bmatrix} \omega_1 \\ \omega_2 \\ \vdots \\ \omega_{n_a} \end{bmatrix} \tag{3}$$

where $f_{uv} = f(\|\boldsymbol{X}_u - \boldsymbol{X}_v\|)$. In a matrix form, **Eq. (3)** can be simply rewritten as $\boldsymbol{Y}_a = \boldsymbol{F}\boldsymbol{\omega}$. This equation has a unique solution $\boldsymbol{\omega} = \boldsymbol{F}^{-1}\boldsymbol{Y}_a$ if and only if all the sample points are different from each other. Therefore, the fill-in values for remaining $n_m$ locations, for which the model responses are missing due to simulation crashes, can be approximated by:

$$\hat{y}(\boldsymbol{X}_j) = \boldsymbol{f}(\boldsymbol{X}_j)\boldsymbol{F}^{-1}\boldsymbol{Y}_a \quad (j = 1,2,\ldots,n_m) \tag{4}$$

To reduce the computational cost and avoid overfitting when building RBF, for each failed simulation at $\boldsymbol{X}_j$ one can choose $k$ non-missing nearest neighbours of that missing value (here we arbitrarily set $k = 100$). Then, a function approximation can be built using these $k$ sample points to approximate that missing value, i.e., in **Eq. (3)**, we set $n_a$ to 100. Moreover, the shape parameter $c$ in the Gaussian kernel function, which is an important factor in the accuracy of the RBF, can be determined using an optimization approach. We use the Nelder-Mead simplex direct search optimization algorithm (Lagarias et al., 1998) to find
an optimal value for $c$ by minimizing the RBF fitting error (for more details see Forrester and Keane (2009) and Kitayama and Yamazaki (2011)).

Note that in general depending on the complexity and dimensionality of the model response surfaces, other types of emulations can be incorporated into our proposed framework. However, for the crash handling problem, it is beneficial to utilize the function approximation techniques that exactly pass through the all sample points (i.e., the response surface
surrogates categorized as "Exact Emulators" in Razavi et al. (2012a)) such as kriging and RBF. This is mainly because most of the DESMs are deterministic, and therefore generate identical outputs/responses given the same set of input factors. In other

words, an exact emulator at any successful sample point $X_k$ (not crashed) reflects our knowledge about the true value of the model's output at that point, i.e., it returns $\hat{y}(X_k)$ without any error.

### 2.3 The utilized GSA frameworks

We illustrate the incorporation of the proposed crash handling methodology into a variogram-based GSA approach called
Variogram Analysis of Response Surfaces (VARS; Razavi and Gupta (2016a)), and a variance-based GSA approach adopted from Saltelli et al. (2008). The VARS framework has successfully been applied to several real-world problems of varying dimensionality and complexity (Sheikholeslami et al., 2017; Yassin et al., 2017; Krogh et al., 2017; Leroux and Pomeroy, 2019; etc.). VARS is a general GSA framework that utilizes directional variograms and covariograms to quantify the full spectrum of sensitivity-related information, thereby providing a comprehensive set of the sensitivity measures called IVARS
(Integrated Variogram Across a Range of Scales) at a range of different "perturbation scales" (Haghnegahdar and Razavi, 2017). Here, we use IVARS-50, referred to as "total-variogram effect", as a comprehensive sensitivity measure since it contains sensitivity analysis information across a full range of perturbation scales.

We utilize the STAR-VARS implementation of the VARS framework (Razavi and Gupta, 2016b). STAR-VARS is a highly efficient and statistically robust algorithm that provides stable results with a minimal number of model runs compared with
other GSA techniques, and thus is suitable for high-dimensional problems (Razavi and Gupta, 2016b). This algorithm employs a star-based sampling scheme, which consists of two steps: (1) randomly selecting star centres in the parameter space, and (2) using a structured sampling technique to identify sample points revolved around the star centres. Due to the structured nature of the generated samples in STAR-VARS, ignorance-based procedures (see section 1.2) cannot be useful in dealing with simulation crashes because deleting sample points associated with crashed simulations will demolish the structure of the entire
sample set. Moreover, to achieve a well-designed computer experiment and sequentially locate star centres in the parameter space, we use the Progressive Latin Hypercube Sampling (PLHS) algorithm. It has been shown that PLHS can grasp the maximum amount of information from the output space with a minimum sample size, while outperforming traditional sampling algorithms (for more details see Sheikholeslami and Razavi, (2017)).

For the variance-based GSA, we calculate the total-effect index (Sobol-TO), which accounts for the impact of any individual
parameter and its interaction with all other parameters, according to the widely used algorithm proposed by Saltelli et al. (2008). This algorithm follows a specific arrangement of randomly generated samples to calculate the sensitivity indices as follows: first, an $n \times 2d$ matrix of independent random numbers is generated (hereafter called "base sample"). Next, by splitting the base sample in half, two new sample matrices, $X^A$ and $X^B$, are built (each of size $n \times d$). Then, to calculate the $i$th sensitivity index $TO_i$, an additional sample matrix of size $n \times d$, $X^{Ci}$ ($i = 1,2,...,d$), is constructed by recombining the
columns of $X^A$ and $X^B$ such that $X^{Ci}$ contains the columns of $X^B$ except the $i$th column which is taken from $X^A$. To build the base sample, we use the Sobol quasi-random sequence. Furthermore, to achieve maximum space-filling properties and to maximize uniformity in the parameter space, for the given sample size, the skip, leap, and scramble operations are applied (for more details see Estrada 2017).

### 3 Case studies

#### 3.1 A conceptual rainfall-runoff model

As an illustrative example, we applied the HBV-SASK conceptual hydrologic model to assess the performance of the proposed crash handling strategies. HBV-SASK is based on Hydrologiska Byråns Vattenbalansavdelning model (Lindström et al., 1997) and was developed by the second author for educational purposes (see Razavi et al., 2019; Gupta and Razavi, 2018). Here, we used HBV-SASK to simulate daily streamflows in the Oldman river basin in Western Canada (Fig. 1) with a watershed area of 1434.73 km$^2$. Historical data is available for periods 1979-2008, from which we estimate average annual precipitation to be 611 mm, and average annual streamflow to be 11.7 m$^3$/s with a runoff ratio of approximately 0.42. HBV-SASK has 12 parameters, of which 10 are perturbed in this study (Table 1).

#### 3.2 A land surface-hydrology model

In the second case study, we demonstrate the utility of the imputation-based methods in crash handling via their application to the GSA of a high-dimensional and much more complex problem. We used the Modélisation Environmentale– Surface et Hydrologie (MESH; Pietroniro et al., (2007)) which is a semi-distributed, highly-parameterized land surface-hydrology modelling framework developed by Environment and Climate Change Canada (ECCC) mainly for large-scale watershed modelling with consideration of cold region processes in Canada. MESH combines the vertical energy and water balance of the Canadian Land Surface Scheme (CLASS, Verseghy, 1991; Verseghy et al., 1993) with the horizontal routing scheme of the WATFLOOD (Kouwen et al., 1993). We encountered a series of simulation failures while assessing the impact of uncertainties in 111 model parameters (see Table A1 in Appendix A) on simulated daily streamflows in Nottawasaga river basin, Ontario, Canada (Fig. 3). For this case study, the drainage basin of nearly 2700 km$^2$ was discretized into 20 grid cells with a spatial resolution of 0.1667 degrees (~15 km). The dominant land cover in the area is cropland followed by deciduous forest and grassland. The dominant soil type in the area is sand followed by silt and clay loam (for more details see Haghnegahdar et al., 2015).

#### 3.3 Experimental setup

In the first case study, for STAR-VARS, we chose to sample 100 star centres (with a resolution of 0.1) from the feasible ranges of parameters (Table 1) using the PLHS algorithm, resulting in 9,100 evaluations of the HBV-SASK model. For the variance-based method, the base sample size was chosen to be 5,000, and thus the model was run 60,000 times. The larger base sample size was selected for the variance-based method to ensure the stability of the algorithm. The Nash-Sutcliffe efficiency criterion on streamflows (NS) was used as the model output for sensitivity analysis. After calculating the NS values, we performed a series of experiments each with a different assumed "ratio of failure" (from 1% to 20%), defined as the percentage of failed parameter sets to the total number of parameter sets. In each experiment, we randomly selected a number of sampled points based on the associated ratio of failure and considered them as simulation failures. Then, we evaluated the performance of the

crash handling strategies in replacing simulation failures during GSA of the HBV-SASK model and compared the results with the case when there are no failures. In addition, we accounted for the randomness in the comparisons by carrying out 50 replicates of each experiment with different random seeds. This allowed us to see a range of possible performances for each strategy and to assess their robustness when crashes occurred at different locations in the parameter space.

In the second case study having 111 parameters, we only tested STAR-VARS with 100 star centres randomly generated using the PLHS algorithm (with a resolution of 0.1), resulting in 100,000 MESH runs. The NS performance metric was used to measure daily model streamflow performance, calculated for a period of three years (October 2003-September 2007) following a one-year model warmup period. Due to various physical and/or numerical constraints inside MESH (or more precisely in CLASS), some combinations of the 111 parameters caused model crashes. Here, approximately 3% of our
simulations failed (3,084 out of 100,000 runs). We applied the proposed crash handling strategies to infill the missing model outcomes in the GSA of the MESH model. The entire set of 100,000 function evaluations of the MESH model would take more than 6 months if we used a single standard CPU core. However, we used the University of Saskatchewan's high-performance computing system to run the GSA experiment in parallel on 160 cores. Therefore, completing all model runs required approximately 32 hours. For this case study, using an Intel® Core™ i7 CPU 4790 3.6GHz desktop PC, the RBF
technique took only 65 seconds to substitute 3,084 crashed runs, while the single NN technique required about 97 seconds to complete the task.

## 4. Numerical results

### 4.1 Results for the HBV-SASK model

According to both of the IVARS-50 and Sobol-TO sensitivity indices, the parameters of the HBV-SASK (when there were no
model crashes) were ranked as follows from the most important to the least important one: {$FRAC$, $FC$, $C0$, $TT$, $alpha$, $K1$, $LP$, $ETF$, $beta$, $K2$}. We assume these rankings and respective sensitivity indices as the "true" values. Based on the dendrogram (Fig. 3) generated by the factor grouping algorithm introduced by Sheikholeslami et al., (2019), we categorized these parameters into three groups with respect to their importance, i.e., {$FRAC$, $FC$, $C0$} are the strongly influential parameters, {$TT$, $alpha$, $K1$} are moderately influential parameters, and {$LP$, $ETF$, $beta$, $K2$} are weakly influential parameters.
Fig. 4, 5, and 6 show the cumulative distribution functions (CDFs) for the 50 independent estimates of IVARS-50, obtained when 1%, 3%, 5%, 8%, 10%, 12%, 15%, and 20% of model runs were deemed to be simulation failures. Overall, the RBF and single NN techniques outperformed the median substitution in terms of closeness to the true GSA results and robustness when crashes happened at different locations of the parameter space.

    As can be seen, by increasing the ratio of failure, the performance of the crash handling strategies, particularly the median
substitution became progressively worse. Note that the median substitution technique resulted in a significant bias manifested through over-estimation of the sensitivity indices for all the parameters. Moreover, Fig. 4 and 6 show that when crashes were substituted using the RBF technique, the STAR-VARS algorithm estimated the sensitivity indices of the most important

parameters {*FRAC*, *FC*, *C0*} (Fig. 4) and less important parameters {*LP*, *ETF*, *beta*, *K2*} (Fig. 6) with high degrees of accuracy and robustness. However, for the moderately influential parameters {*TT*, *alpha*, *K1*} in Fig. 5, its performance degraded (i.e., the CDFs are wider in Fig. 5). The respective results using the variance-based algorithm are presented in Fig. B1, B2, and B3 for strongly influential, moderately influential, and weakly influential parameters, respectively (see Appendix B). Because our proposed approach for crash handling is GSA-method-free, we observed a similar performance when using the variance-based algorithm. In other words, the RBF effectively handled the crashes and produced reasonable sensitivity analysis results compared to the NN and median substitution techniques.

More importantly, as the number of crashes increases, rankings of the parameters in terms of their importance may change. Fig. 7 and 8 show the number of times out of 50 independent runs that the rankings of the parameters were equal to the "true" ranking for the STAR-VARS and variance-based GSA algorithms. In all 50 runs, regardless of the number of model crashes, the rankings obtained by the STAR-VARS using the RBF technique were equal to the "true" ranking, indicating a high degree of robustness in terms of parameter ranking. The performance of the single NN slightly decreased when the crash percentage were more than 15%, while the STAR-VARS algorithm wrongly determined the rankings in more than 50% percent of the replicates using median substitution technique (see Fig. 7c and d). This highlights that the rankings can be estimated much more accurately than the sensitivity indices in the presence of simulation crashes. In addition, it can be seen that while the RBF-based strategy performed perfectly in this example, the performance of the single NN technique was comparably well (Fig. 7). However, for the variance-based technique, only the rankings of the most important parameters were equal to the "true" ranking, regardless of the number of model crashes and the utilized crash handling strategy (Fig. 8). Moreover, the performance reduction of the single NN technique was higher when the variance-based method was employed. In fact, the variance-based algorithm wrongly estimated the rankings in more than 30% percent of the replicates using the single NN technique when the ratio of failure was 15% (Fig. 8c) and 20% (Fig. 8d).

Finally, Fig. 9 presents the performance of the single NN (Fig. 9a and c) and RBF (Fig. 9b and d) strategies in approximating the fill-in values for the missing responses when 5% (upper panel) and 20% (bottom panel) of the HBV-SASK simulations were deemed failures. As shown, the RBF outperformed single NN technique in terms of closeness to the true NS values. For example, having 20% of the model runs failed, the linear regression has an $R^2$ value of 0.834 when single NN was used, while the RBF strategy achieved a linear regression with an $R^2$ value of 0.996. In fact, the results of the RBF strategy are almost unbiased, as the linear regression plotted on Fig. 9b and d is very close to the ideal (1:1) line.

**4.2 Results for the MESH model**

We demonstrate the GSA results of the MESH model by categorizing the 111 parameters of the model into three groups as shown in Fig. 10 (for more details on grouping see Sheikholeslami et al. (2019)). Fig. 11-13 present the sensitivity analysis results obtained by the STAR-VARS algorithm for the MESH model, when we applied different crash handling strategies. These groups are labelled according to their importance, i.e., Group 1 (Fig. 11) contains the strongly influential parameters,

while parameters in Group 2 (Fig. 12) are moderately influential, and Group 3 (Fig. 13) is the group of weakly influential parameters.

The four most influential parameters in Group 1 are *SDEPC* and *DRNC* ("C" stands for crops), controlling the water storage and water movement in the soil, *WFR22* (river channel routing), and *ZSNL* (snow cover fraction). As shown in Fig. 11 (upper panel), the sensitivity indices associated with these parameters are almost similar regardless of the employed crash handling technique. As discussed in our failure analysis (see Section 5.1), we also identified three of these parameters (i.e., *SDEPC*, *DRNC*, and *ZSNL*) responsible for some of the model crashes. In other words, those parameters that strongly contribute to the variability of the MESH model output can also be convicted of model crashes. To enhance future development and application of the MESH model, more efforts should be directed at better understanding the functioning of these parameters and their effects acting individually or in combination with other parameters over their entire range of variations.

For the other 15 influential parameters in Group 1 (Fig. 11, bottom panel), there is general agreement between these three crash handling techniques about the sensitivity indices calculated by the STAR-VARS except for the parameter *ROOTC*, which defines the annual maximum rooting depth of a vegetation category. The RBF and median substitution methods give more importance to *ROOTC* compared to the single NN technique. It is noteworthy that the oversaturation of the soil layer, which can cause many model runs to fail, is subject to the interaction between *ROOTC* and *SDEPC*.

Fig. 12 illustrates the sensitivity indices for the moderately influential parameters (i.e., Group 2). For all these 78 parameters, the sensitivity analysis results were highly dependent on the chosen crash handling strategy. As can be seen, the sensitivity indices associated with the median substitution and RBF techniques are higher than those obtained by the single NN technique (this difference is considerable for the parameters in the upper and lower subplots than those in the middle subplot).

Finally, the results of the sensitivity analysis for the weakly or non-influential (Group 3) parameters of the MESH model are plotted in Fig. 13. The STAR-VARS algorithm identified these parameters as weakly influential (very low IVARS-50 values) using the proposed crash handling techniques. However, the associated sensitivity indices obtained by the RBF imputation method are two orders of magnitude larger for the parameters in the left panel (Fig.13 (a, c)) and about four orders of magnitude larger for the parameters in the right panel (Fig. 13 (b, d)) compared to those obtained by the single NN and median substitution methods.

It is important to note that in high-dimensional DESMs, when the number of parameters is very large, the estimation of sensitivity indices is likely not robust to sampling variability. On the other hand, parameter ranking (order of relative sensitivity) is often more robust to sampling variability and converges more quickly than factor sensitivity indices (see e.g., Vanrolleghem et al., 2015; Razavi and Gupta, 2016b; Sheikholeslami et al., 2019). To investigate how different crash handling strategies can affect the ranking of the model parameters in terms of their importance, Fig. 14 compares the rankings obtained by the RBF, single NN, and median substitution techniques.

As shown in Fig. 14a, the single NN and median substitution techniques resulted in almost similar parameter rankings for the strongly influential (Group 1) and weakly influential (Group 3) parameters, while for moderately influential parameters (Group 2) the rankings are significantly different. Meanwhile, the RBF and median substitution techniques yielded very

distinctive rankings except for the strongly influential parameters (Fig. 14b). Furthermore, Fig. 14c indicates that the single NN and RBF methods provided similar rankings for the influential parameters.

A closer examination, however, reveals that rankings can be contradictory for some of the parameters when using different crash handling strategies (see Fig. 14 d-f). For example, consider the soil moisture suction coefficient for crops (*PSGAC*)
which is used in calculation of the stomatal resistance in the evapotranspiration process of the MESH (for more details see Fisher et al., 1981; Choudhury and Idso 1985; Verseghy, 2012). As can be seen, according to the RBF method, *PSGAC* is one of the weakly influential parameters (ranked 5th) (note that a ranking of 1 means the least influential, while ranking of 111 means the most influential parameter), while using the single NN it is determined to be one of the moderately influential parameters (ranked 43rd). In contrast, it is one of the strongly influential parameters based on the median substitution (ranked
83rd). However, in a comprehensive study of the MESH model using various model configurations and different hydroclimatic regions in Eastern and Western Canada, Haghnegahdar et al. (2017) found that *PSGAC* is one of the least influential parameters considering three model performance criteria with respect to high flows, low flows, and total flow volume of the daily hydrograph. As another example, consider *ZPLS7* (maximum water ponding depth for snow-covered areas) and *ZPLG7* (maximum water ponding depth for snow-free areas) which are used in surface runoff algorithm of the MESH (i.e., PDMROF).
The single NN and median substitution methods both ranked *ZPLS7* as second and *ZPLG7* as third least influential parameters, whereas the RBF ranked them as 61 and 45 (i.e., moderately influential) which is in accordance with the results reported by Haghnegahdar et al. (2017).

## 5. Discussion

### 5.1 Potential causes of failure in the MESH

Our further investigations of the MESH model revealed at least two possible causes responsible for many of the simulation failures, i.e., the threshold behaviour of some parameters and oversaturation of the soil layers. For example, the threshold behaviour of the *ZSNL* (the snow depth threshold below which snow coverage is considered less than 100%) might cause many model crashes. When *ZSNL* was relatively large, it resulted in calculation of overly thick snow columns inside the model, violating the energy balance constraints and triggering a simulation abort. This situation became more severe when the
calculated snow depth was larger than the maximum vegetation height(s). Fig. 15 (left column) shows the scatterplots of the *ZSNL* values sampled from the feasible ranges for all model simulations used for GSA of the MESH, with failed designs marked by red dots.

Furthermore, from our analysis we found that the oversaturation of the soil layer may happen especially at lower values of the soil permeable depth (*SDEP*) and also when it becomes less than the maximum vegetation rooting depth (*ROOT*). The
situation is more severe when the soil drainage index (*DRN*) is reduced. These interactions can collectively cause a thinner soil column for water storage and movement that now has a lower chance for transpiration and drainage, thereby resulting in over accumulation of the water beyond the physical limits set for the soil in the model. Fig. 15 (right column) displays the

pairwise scatterplots of the *SDEP*, *ROOT*, and *DRN*. To avoid model crashes, it is necessary to ensure that *SDEP* and *ROOT* values are not unrealistically low and that their values and/or their ranges are assigned as accurately as possible using the available data.

As can be seen from Fig. 15, very high values of parameters *DRNC* and *SDEPC* can also cause simulation crashes, while these crashes were happened at lower values of *ZSNL7*. Note that from these 2-dimensional projections of the 111-dimensional parameter space of the MESH no general conclusions can be drawn. This even becomes more complicated when noticing some isolated crashes in regions where most of the simulations were successful. Furthermore, as shown in Fig. 15, there are considerable overlaps between successful simulations and crashed ones in the feasible ranges of parameters. For example, there are many crashed simulations when *DRNC* was sampled from [3.5-4], at the same time a high density of successful simulations can also be observed in the same range. This indicates that locating regions of parameter space responsible for crashes is difficult, if not impossible, and necessitates analysing the MESH's response surface throughout a high-dimensional parameter space.

## 5.2 The role of sampling strategies in handling model crashes

Due to the extremely large parameter space ($X$) of high-dimensional DESMs, it may require many properly distributed sample points ($X_s$) to generate/explore a full spectrum of model behaviors such as simulation crashes, discontinuities, stable regions, optima, etc. Together with the computationally intensive nature of DESMs, this issue can make both non-substitution procedures and imputation-based methods (those proposed in the present study) very costly in dealing with crashes, if not impractical. It is important to note that the sample size in GSA studies should not only be determined based on the available computational budget but also considerations of GSA stability and convergence. Therefore, it is of vital importance to monitor and evaluate the convergence rate of the GSA algorithms. Strategies introduced by Nossent et al. (2011), Sarrazin et al. (2016), and more recently by Sheikholeslami et al. (2019) enable users to diagnose the convergence behaviour of the GSA algorithms.

Because the non-substitution procedures rely on constructing a statistical model based on observed crashes to predict and avoid them in the follow-up experiments, they need a good coverage of the domain to attain a reliable statistical model. This issue also challenges the use of imputation-based methods. For example, in the NN techniques (both single and *k*-NN) one major concern is that the sparseness of sample points may affect the quality of the results. In regions of the parameter space where the sample points are sparsely distributed, distances to nearest neighbours can be relatively large, leading to choosing physically incompatible neighbours. Moreover, in response surface modelling-based techniques, building an accurate and robust function approximation directly depends on the utilized sampling strategy and how dense mappings between parameter and output spaces are (see, e.g., Jin et al., 2001; Mullur and Messac, 2006; Zhaou and Xue, 2010).

A crucial consideration in the use of any sampling strategy is the exploration ability of that strategy (i.e., space-filling ability), which significantly influences the effectiveness of the utilized crash handling approach. When having this feature enabled (i.e., exploration), the non-substitution procedures can reliably identify implausible regions in the entire parameter space, meaning that the sample set is not confined to only a limited number of regions. Furthermore, it can notably improve

the predictive accuracy of the response surface modelling-based methods (Crombecq et al., 2011). Exploration requires sample points to be evenly spread across the entire parameter space to ensure that all regions of the domain are equally explored, and thus sample points should be located almost equally apart. This feature rectifies the problems relating to the distances between sample points when using NN techniques since in space-filling designs these distances should be as evenly distributed as possible.

Given this, regardless of the chosen method for solving simulation crash problem in GSA, it is advisable to spend some time up front to find an optimal sample set before submitting it for evaluation to the computationally expensive DESMs. It is, therefore, necessary to prudently use improved sampling algorithms such as Progressive Latin Hypercube Sampling (PLHS; Sheikholeslami and Razavi (2017)), K-extended Latin Hypercubes (k-extended LHCs; Williamson (2015)), or Sequential Exploratory Experimental Design (SEED; Lin (2004)). Generally, these sampling techniques optimize some characteristics of the sample points such as sample size, space-filling properties, projective properties, etc.

We conclude this section by highlighting a point that should receive careful attention when applying the substitution-based methods in handling model crashes. In addition to the numerical artefacts in simulation models, some combinations of parameter values, which may not be physically justified, can also lead to simulation failures. As a result, there is risks that substituting data for these crashed runs contaminate the assessment of parameter importance. Preventing this type of risks requires knowledge about the reasonable parameter ranges in DESMs. This type of crashes can be significantly reduced by selecting plausible ranges of parameters based on physical knowledge or information of the problem (a process referred to as "parameter space refinement" (see e.g., Li et al., 2019; Williamson et al., 2013)). However, DESMs often consist of many interacting, uncertain parameters, and therefore very little may be known a priori about the implausible regions of the parameter space.

## 6 Conclusion

Understanding the complex physical processes in Earth and environmental systems and predicting their future behaviours rely routinely on high-dimensional, computationally expensive models. These models are often involved in the processes of model calibration, and/or uncertainty and sensitivity analysis. If a simulation failure/crash occurs at any of these processes, these models will stop functioning, and thus need user intervention. Generally, there are many reasons for failure of a simulation in models, including the use of inconsistent integration time steps or grid resolutions, lack of convergence, and inadequate threshold behaviours in models. Determining whether these "defects" exist in the utilized numerical schemes or they are programming bugs can only be done through analysing a high-dimensional parameter space and characterizing implausible regions responsible for crashes. This imposes a heavier computational burden on analysts. More importantly, every "crashed" simulation can be very demanding in terms of computational cost for global sensitivity analysis (GSA) algorithms because they can prevent the completion of the analysis and introduce ambiguity into the GSA results.

These challenges motivated us to implement missing data imputation-based strategies for handling simulation crashes in the GSA context. These strategies involve substituting plausible values for the failed simulations in the absence of a priori knowledge regarding the nature of the failures. Here, our focus was to find simple yet computationally frugal techniques to palliate the effect of model crashes on the GSA of Dynamical Earth System Models (DESMs). Thus, we utilized three techniques, including median substitution, single nearest neighbour, and emulation-based substitution (here we used radial basis functions as a surrogate model) to fill in a value for the failed simulations using available information and other non-missing model responses. The high efficiency of our proposed substitution-based approach is of prominent importance, particularly when dealing with GSA of the computationally expensive models mainly because our proposed approach does not need repeating the entire experiment.

We compared the performance of our approach in GSA of two modelling case studies in Canada, including a 10-parameter HBV-SASK conceptual hydrologic model and a 111-parameter MESH land surface-hydrology model. Our analyses revealed that:

- Overall, the emulation-based substitution can effectively handle the simulation crashes and produce promising sensitivity analysis results compared to the single nearest neighbour and median substitution techniques.
- As expected, the performance of the proposed methods deteriorates as the ratio of failures increases. The rate of degradation depends on the number of model parameters (dimensionality of the parameter space).
- We observed in our experiments that the utilized crash handling strategy (i.e., median substitution, single NN, and RBF) has minimum influence on the rankings of the strongly and weakly influential parameters identified by the GSA algorithms, while for the moderately influential parameters, different strategies yielded different rankings.

Furthermore, we conducted a failure analysis for the second case study (MESH model) and identified some parameters that seem to be frequently causing model failures. Such analyses are helpful and much needed to improve the fidelity and numerical stability of the DESMs and may constitute a promising avenue of research. In doing so, applying other advanced methods (see e.g., Lucas et al. (2013)) can be beneficial to diagnose existing defects of the complex models.

Future work should include extending the proposed crash handling approach to time-varying sensitivity analysis of the DESMs because a comprehensive GSA requires a full consideration of the dynamical nature of the models. Our proposed approach can be integrated with any time-varying sensitivity analysis algorithm, for example, with the recently developed Generalized Global Sensitivity Matrix (GGSM) method (Gupta and Razavi, 2018; Razavi and Gupta, 2019). This further helps understanding the temporal variation of the parameter importance and model behaviour. Finally, another possible future direction is to apply and test other types of emulation techniques such as kriging and support vector machine in handling model crashes.

**Code Availability.**

The MATLAB codes for the proposed crash handling approach and the HBV-SASK model are included in the VARS-TOOL software package, which is a comprehensive, multi-algorithm toolbox for sensitivity and uncertainty analysis (Razavi et al., 2019). VARS-TOOL is freely available for non-commercial use and can be downloaded from http://vars-tool.com/. The most recent version of the MESH model can be downloaded from https://wiki.usask.ca/display/MESH/Releases. Additional data and information are available upon request from the authors.

**Appendix A: Parameters of the MESH model**

Parameters of the MESH model and their corresponding groups are listed in Table A. The description of parameters and their feasible ranges can be found in Haghnegahdar et al. (2017).

**Appendix B: Performance of the crash handling strategies in sensitivity analysis of the HBV-SASK model using the variance-based algorithm**

**Author contributions.**

All authors contributed to conceiving the ideas of the study. RS and SR designed the method and experiments. The simulations for the first case study were carried out by RS. AH developed the second case study and performed the MESH simulations. RS developed the MATLAB codes for the proposed crash handling approach and conducted all the experiments. RS wrote the manuscript with contributions from SR and AH. All authors contributed to the interpretation of the results, structuring and editing of the paper at all stages.

**Competing interests.**

The authors declare that they have no conflict of interest.

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

**(a) Saskatchewan River Basin**

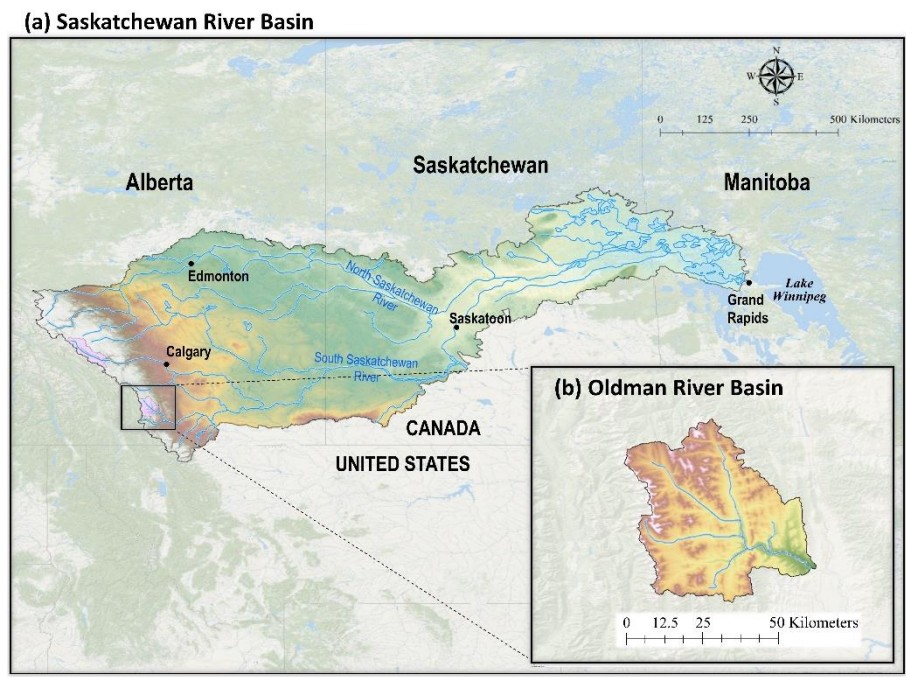

Figure 1: Oldman river basin (b) located in the Rocky Mountains in Alberta, Canada, flows into the Saskatchewan River Basin (a).

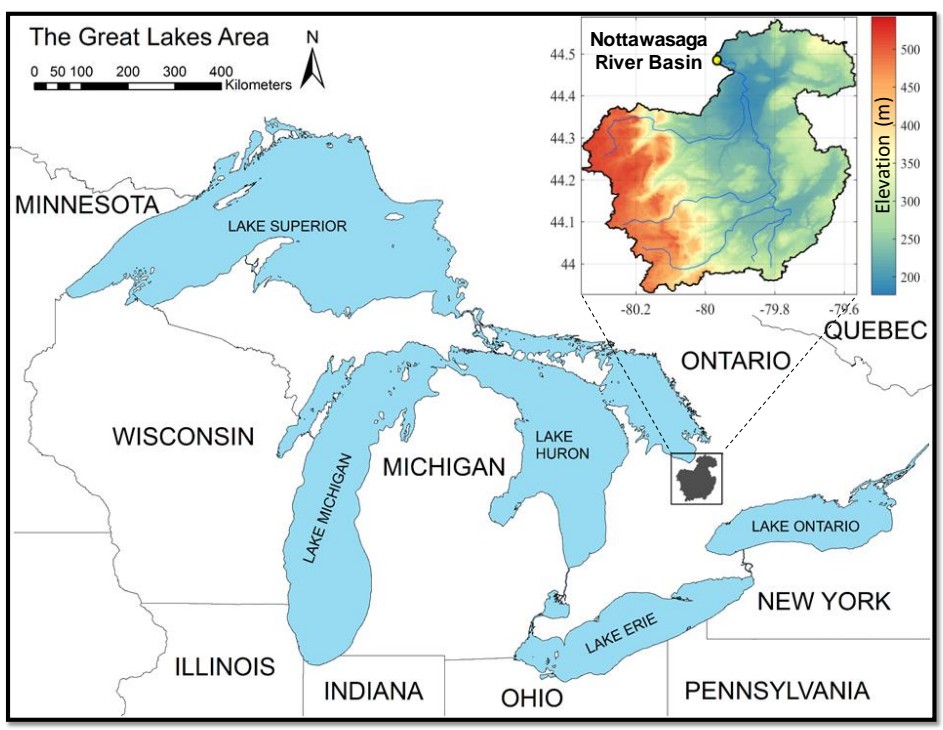

**Figure 2: Nottawasaga river basin in in Southern Ontario, Canada.**

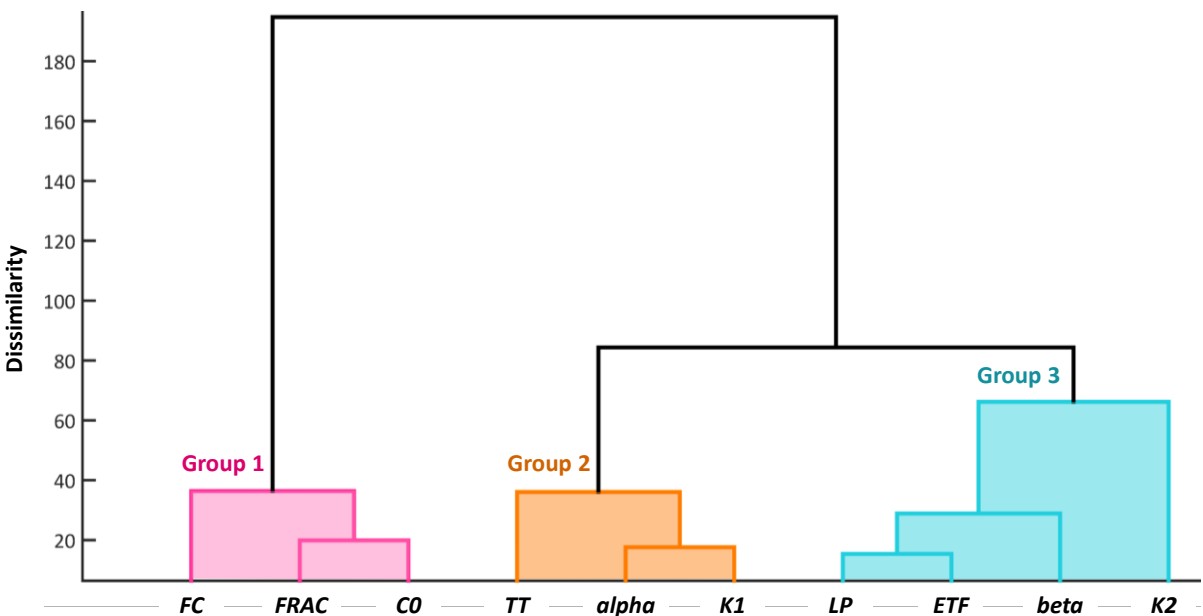

**Figure 3: Grouping of the 10 parameters of the HBV-SASK model when applied on the Oldman River Basin. The parameters are sorted from the most influential (to the left) to the least influential (to the right).**

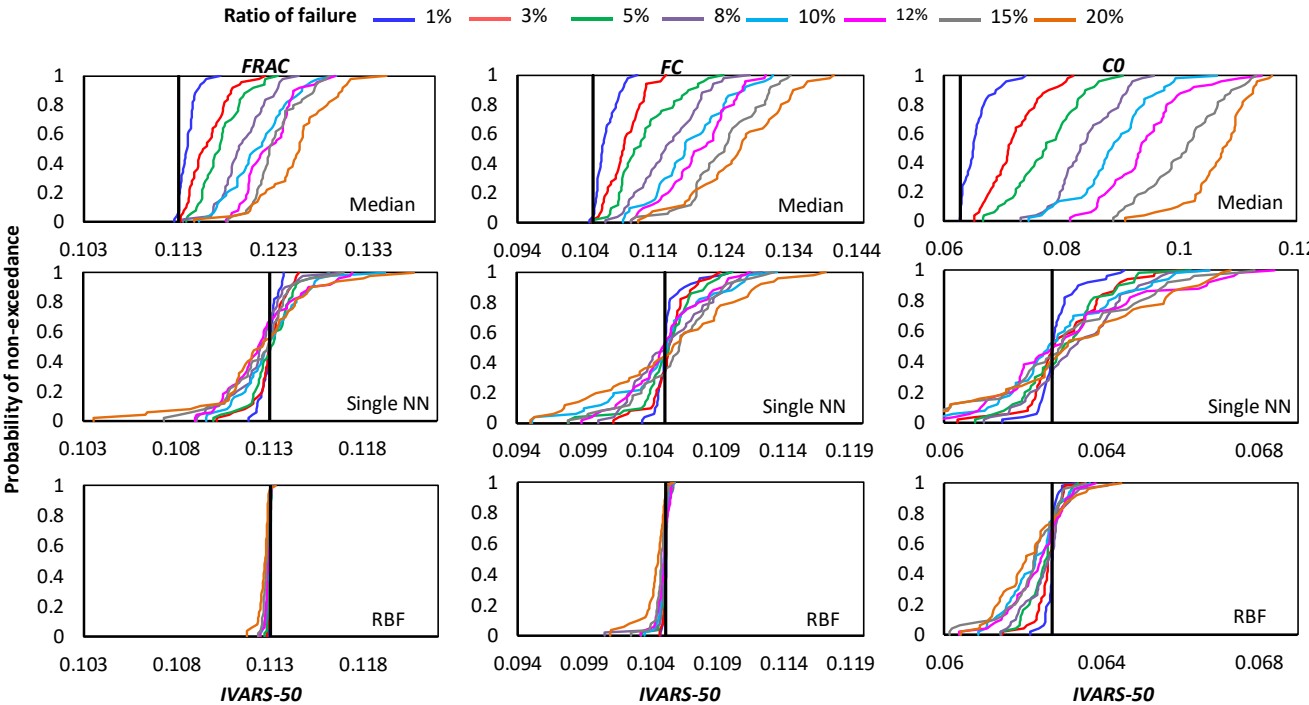

**Figure 4: Comparison of the proposed crash handling strategies in sensitivity analysis of the HBV-SASK model using the STAR-VARS algorithm for different ratios of failures. The CDFs of the sensitivity indices for strongly influential parameters {*FRAC*, *FC*, *C0*} are compared in this plot. The vertical line (solid black) on each subplot represents the corresponding "true" sensitivity index obtained when there were no failures.**

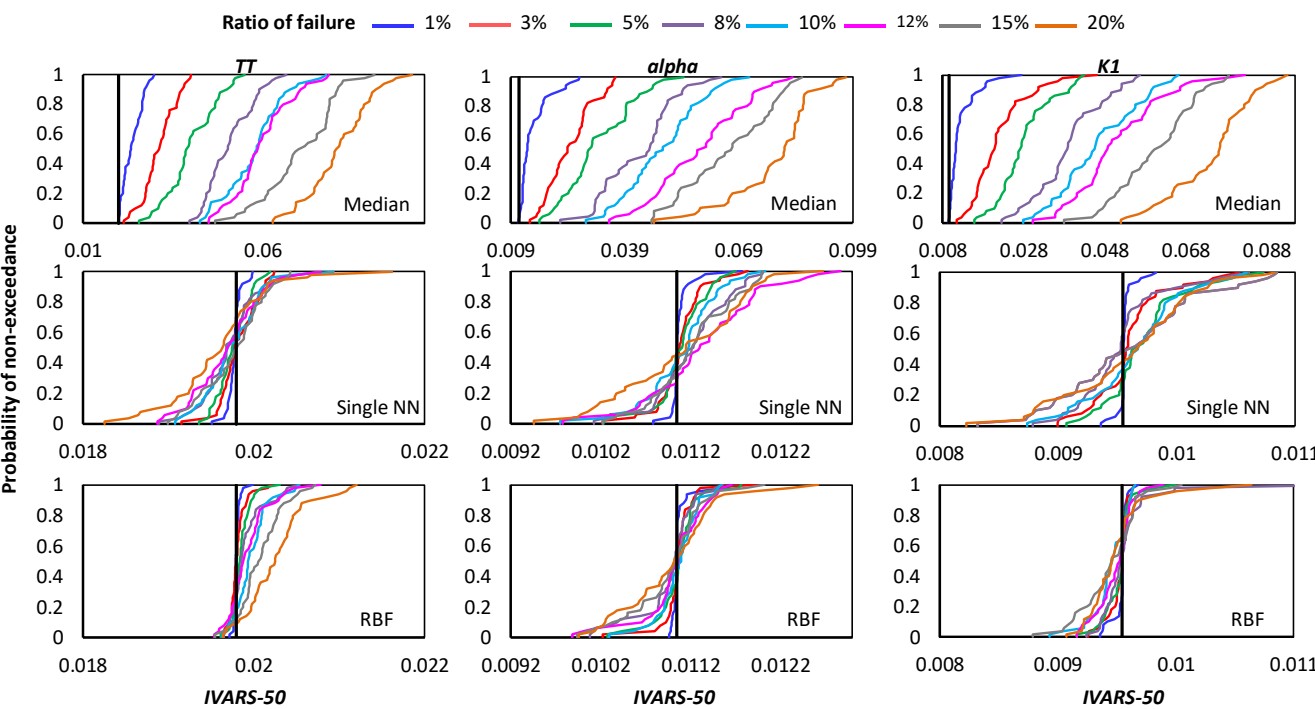

**Figure 5: Comparison of the proposed crash handling strategies in sensitivity analysis of the HBV-SASK model using the STAR-VARS algorithm for different ratios of failures. The CDFs of the sensitivity indices for moderately influential parameters {*TT*, *alpha*, *K1*} are compared in this plot. The vertical line (solid black) on each subplot represents the corresponding ''true'' sensitivity index obtained when there were no failures.**

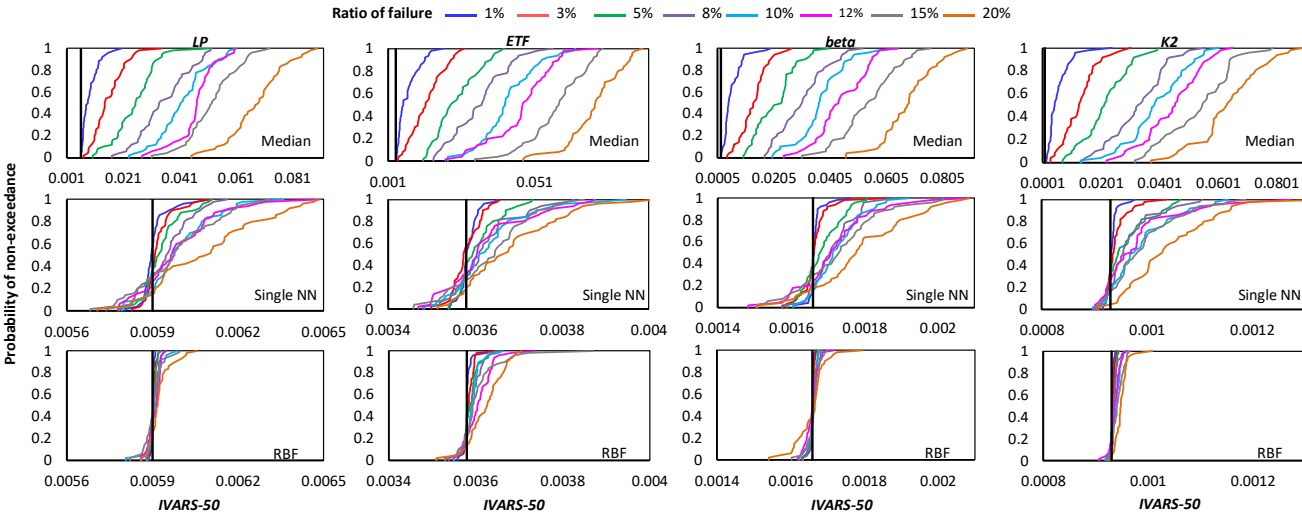

**Figure 6: Comparison of the proposed crash handling strategies in sensitivity analysis of the HBV-SASK model using the STAR-VARS algorithm for different ratios of failures. The CDFs of the sensitivity indices for weakly influential parameters (*LP*, *ETF*, *beta*, *K2*) are shown in this plot. The vertical line (solid black) on each subplot represents the corresponding "true" sensitivity index obtained when there were no failures.**

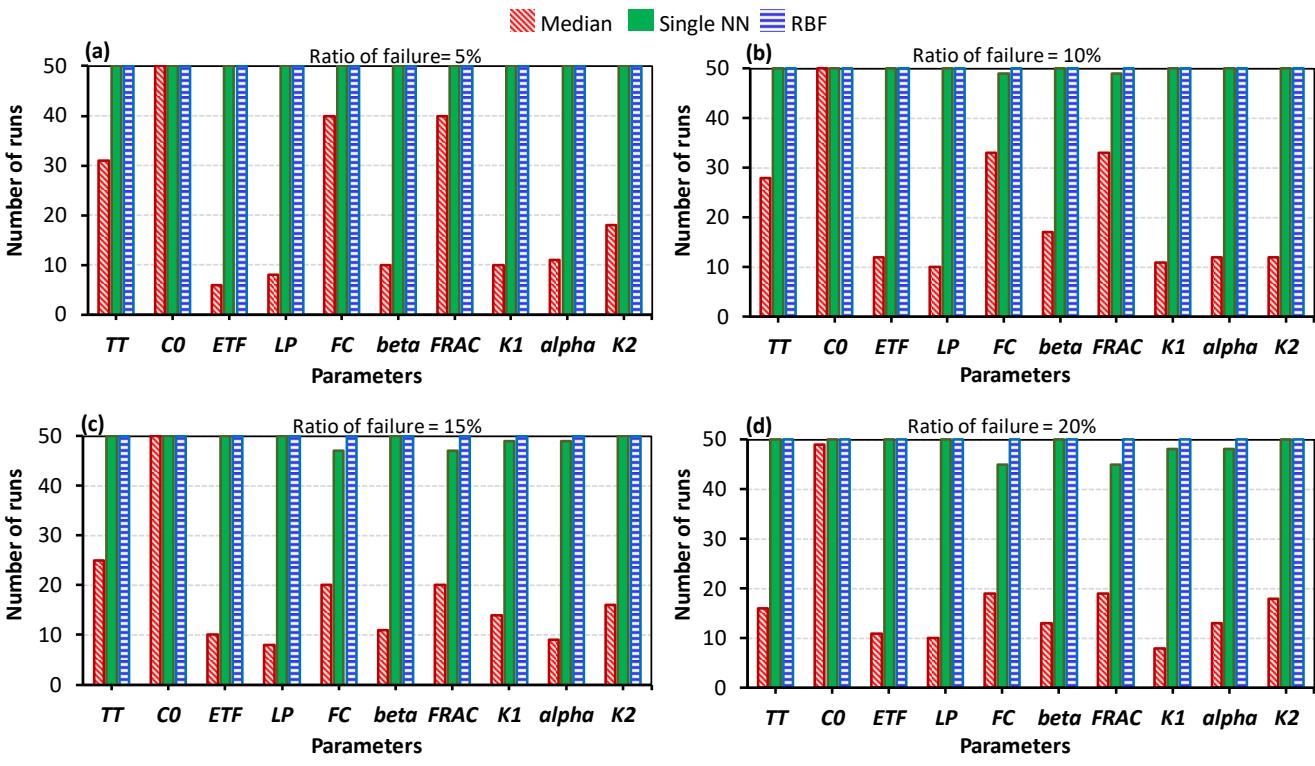

**Figure 7: Comparison of the crash handling strategies in estimating the parameter rankings for HBV-SASK model using the STAR-VARS algorithm when the ratio of failure was (a) 5%, (b) 10%, (c) 15%, and (d) 20%. The *y*-axis in each subplot shows the number of times out of 50 replicates that the rankings of the parameters were equal to the true ranking.**

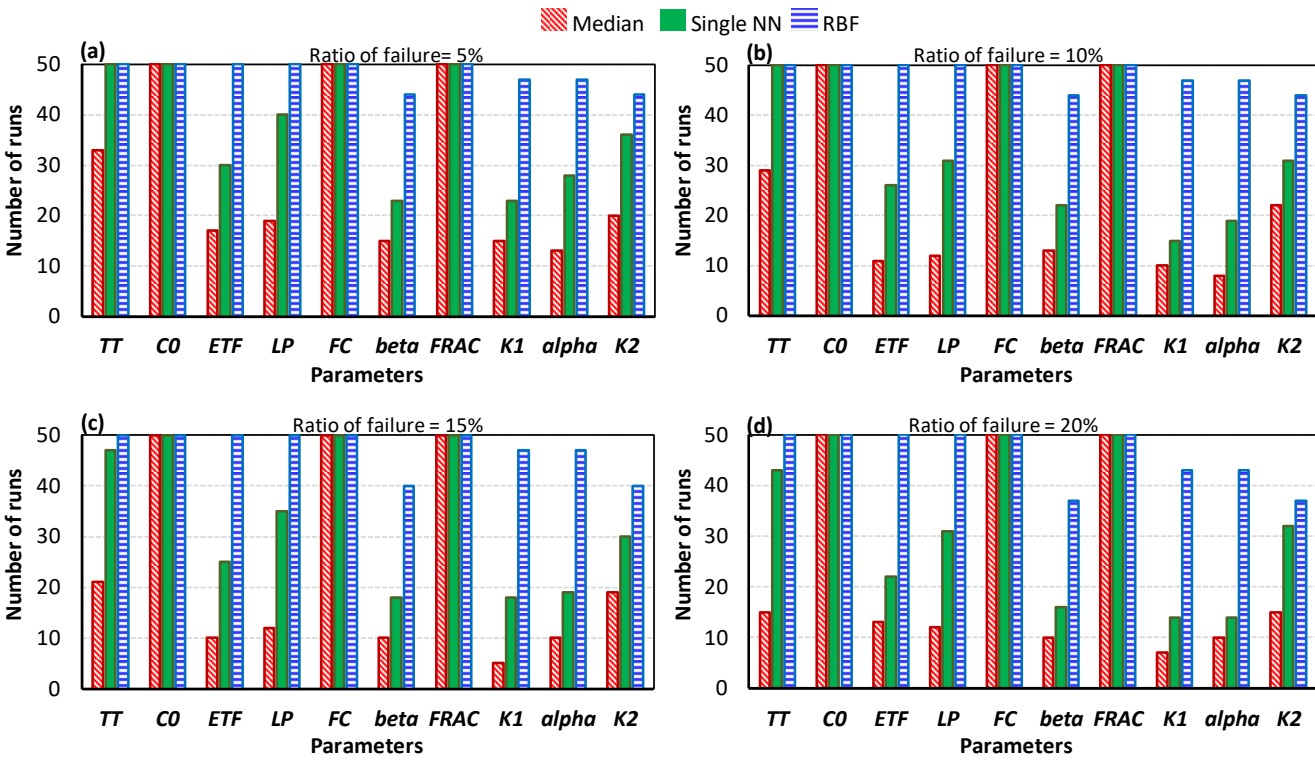

**Figure 8: Comparison of the crash handling strategies in estimating the parameter rankings for HBV-SASK model using the variance-based algorithm when the ratio of failure was (a) 5%, (b) 10%, (c) 15%, and (d) 20%. The *y*-axis in each subplot shows the number of times out of 50 replicates that the rankings of the parameters were equal to the true ranking.**

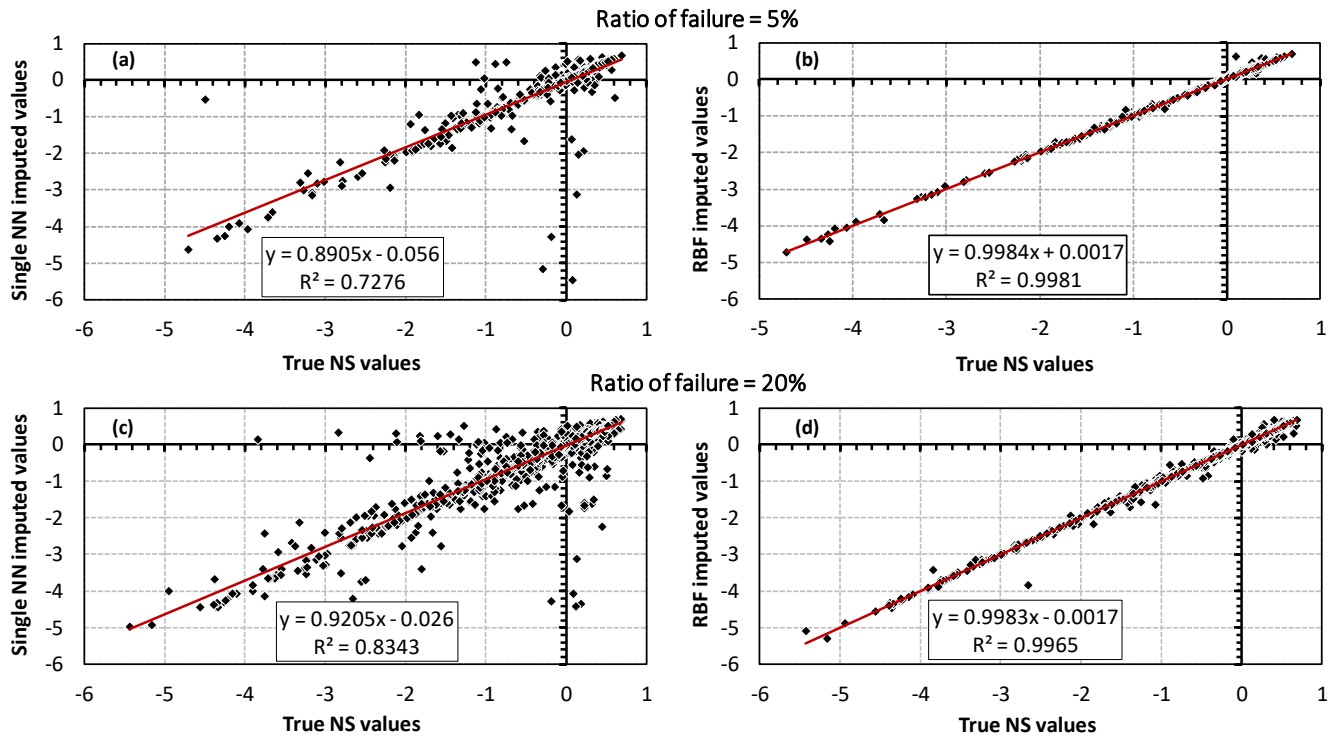

**Figure 9: Scatter plots of the true NS values versus the imputed NS values when the ratio of failure was 5% (top panel) and 20% (bottom panel) for the HBV-SASK model. The accuracy of the crash handling strategies is demonstrated in subplots (a) and (c) for the single NN method and in subplots (b) and (d) for the RBF method. These results belong to one arbitrarily chosen replicate out of 50 independent runs.**

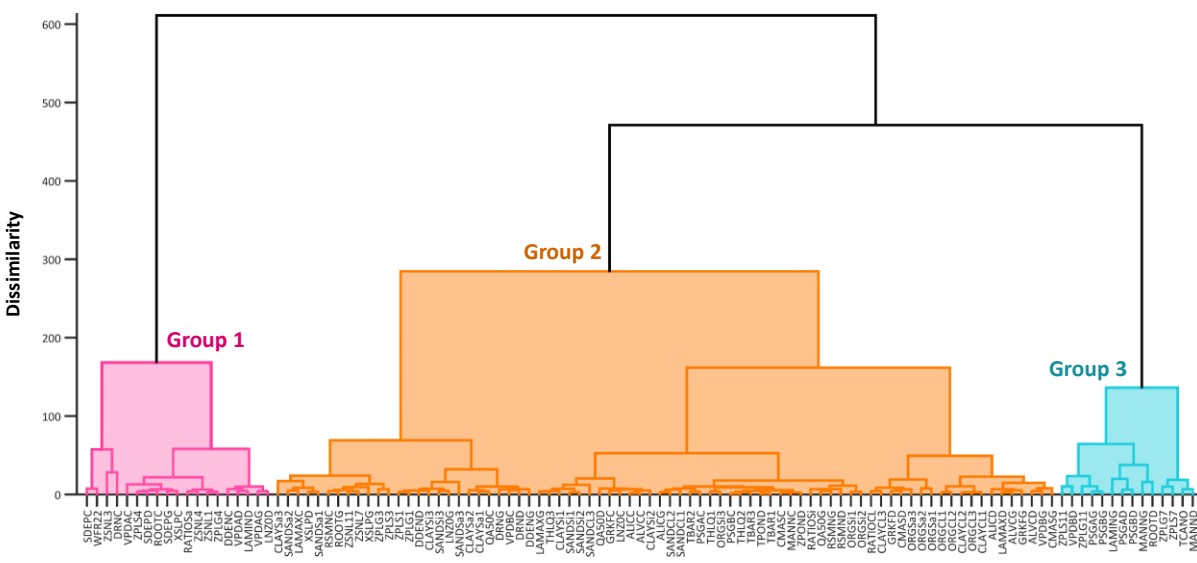

**Figure 10: Grouping of the 111 parameters of the MESH model. The parameters are sorted from the most influential (to the left) to the least influential (to the right). This grouping is based on the results of the RBF method.**

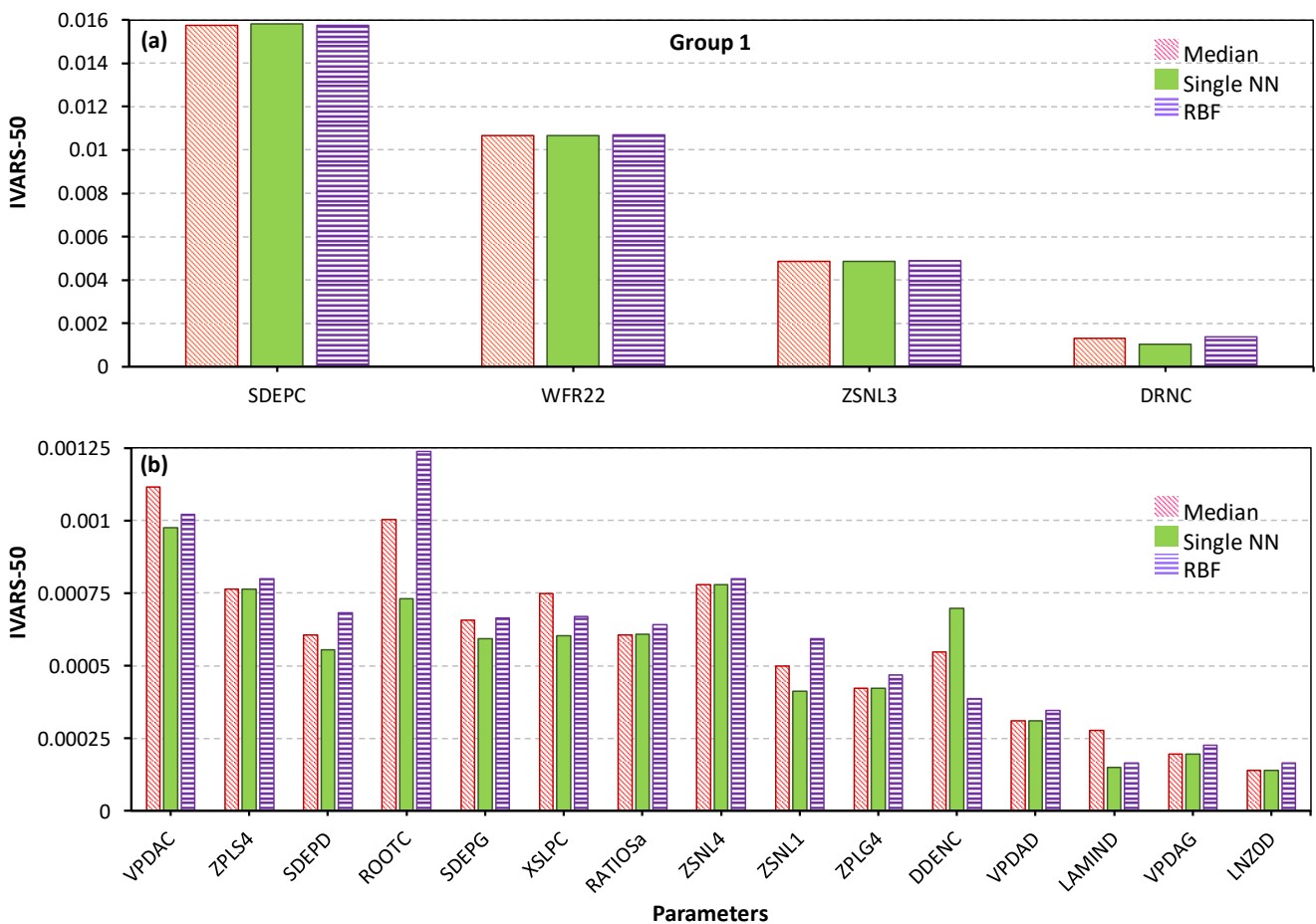

**Figure 11: Sensitivity analysis results of the MESH model using different crash handling strategies for the most influential parameters. To better illustrate the results, the highly influential parameters in Group 1 (see Figure 10) are separately shown in two subplots (a) and (b).**

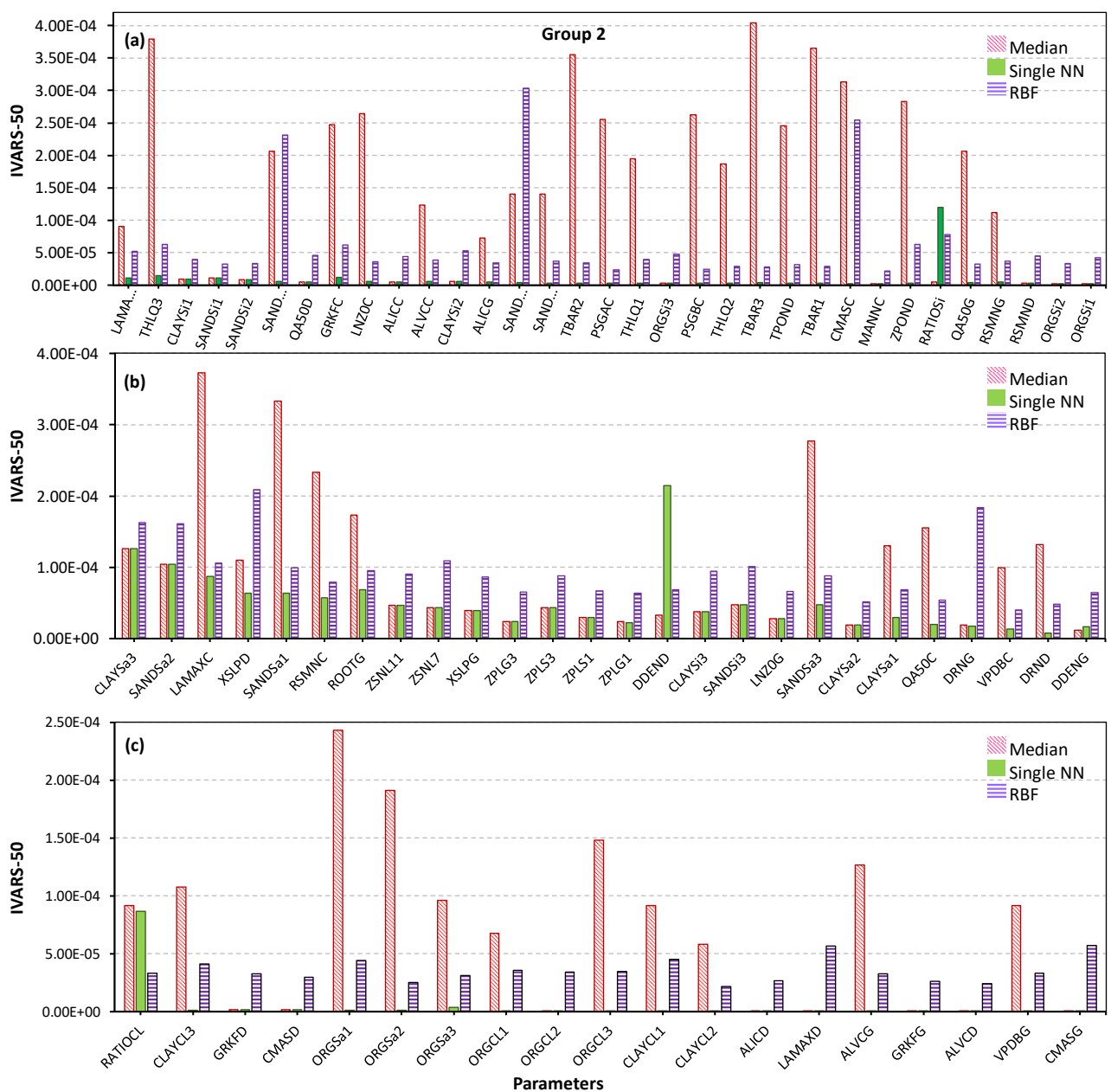

**Figure 12: Sensitivity analysis results of the MESH model for moderately influential parameters using different crash handling strategies. To better illustrate the results, the moderately influential parameters in Group 2 (see Figure 10) are separately shown in three subplots (a), (b), and (c).**

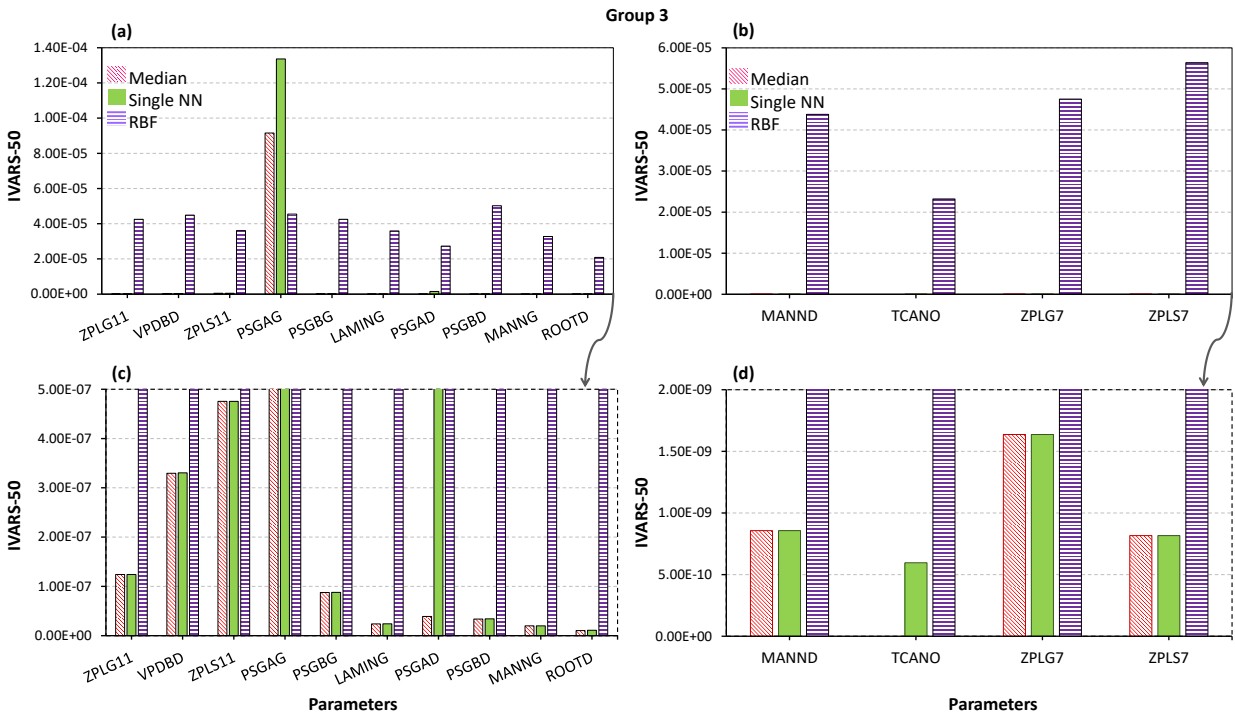

**Figure 13: Sensitivity analysis results of the MESH model using different crash handling strategies for weakly/non- influential parameters in Group 3 (see Figure 10). The bottom panel (c and d) shows a zoom-in of the top subplots for very small values on the vertical axis.**

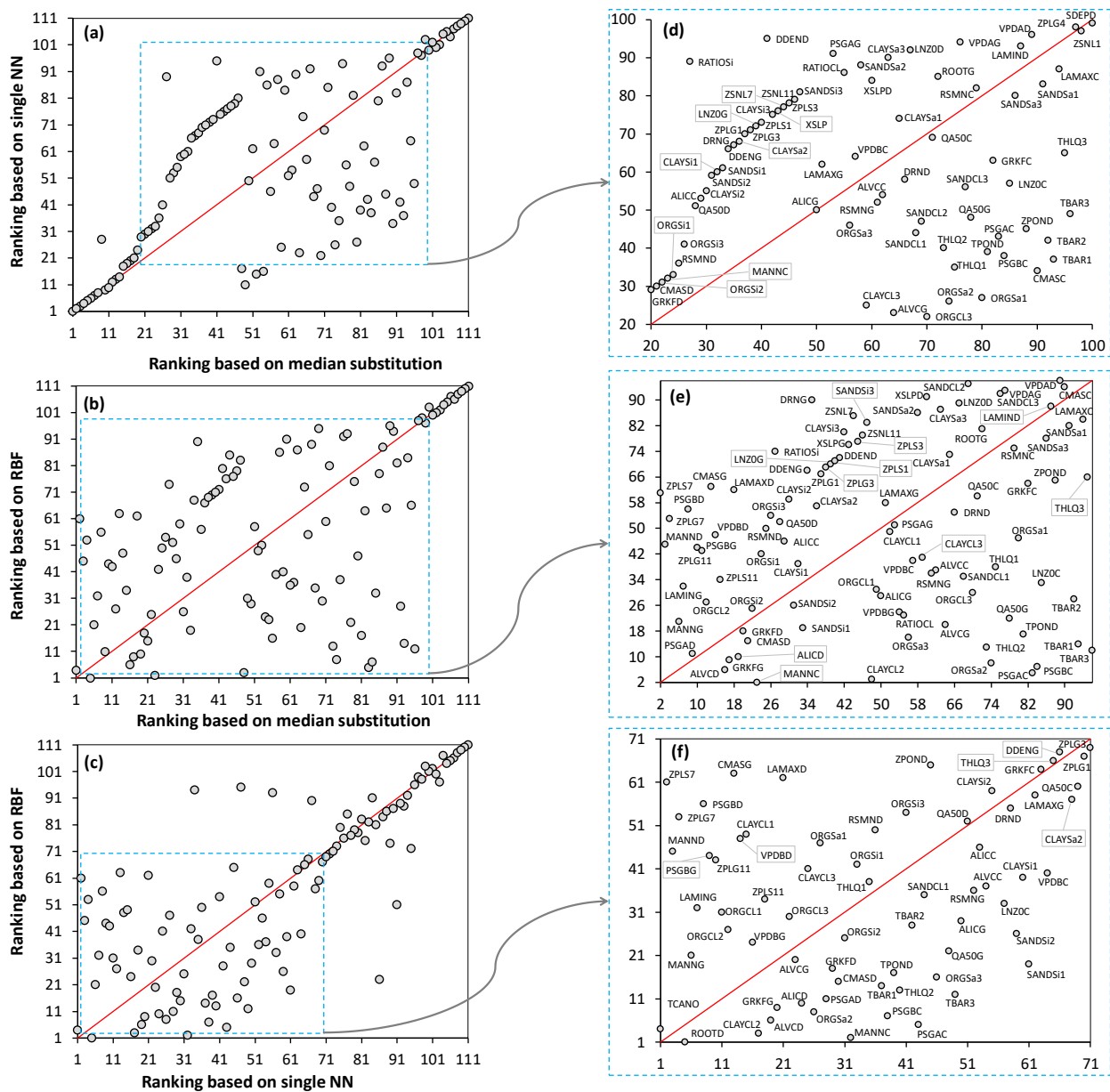

5  **Figure: 14. Comparing rankings of the MESH model parameters obtained by different crash handling strategies using the STAR-VARS algorithm. Subplots (d), (e), and (f) (right column) show a zoom-in of the subplots (a), (b), and (c) (left column), respectively. The red line is the ideal (1:1) line. Note that a ranking of 1 represents the least influential and a ranking of 111 represents the most influential parameter.**

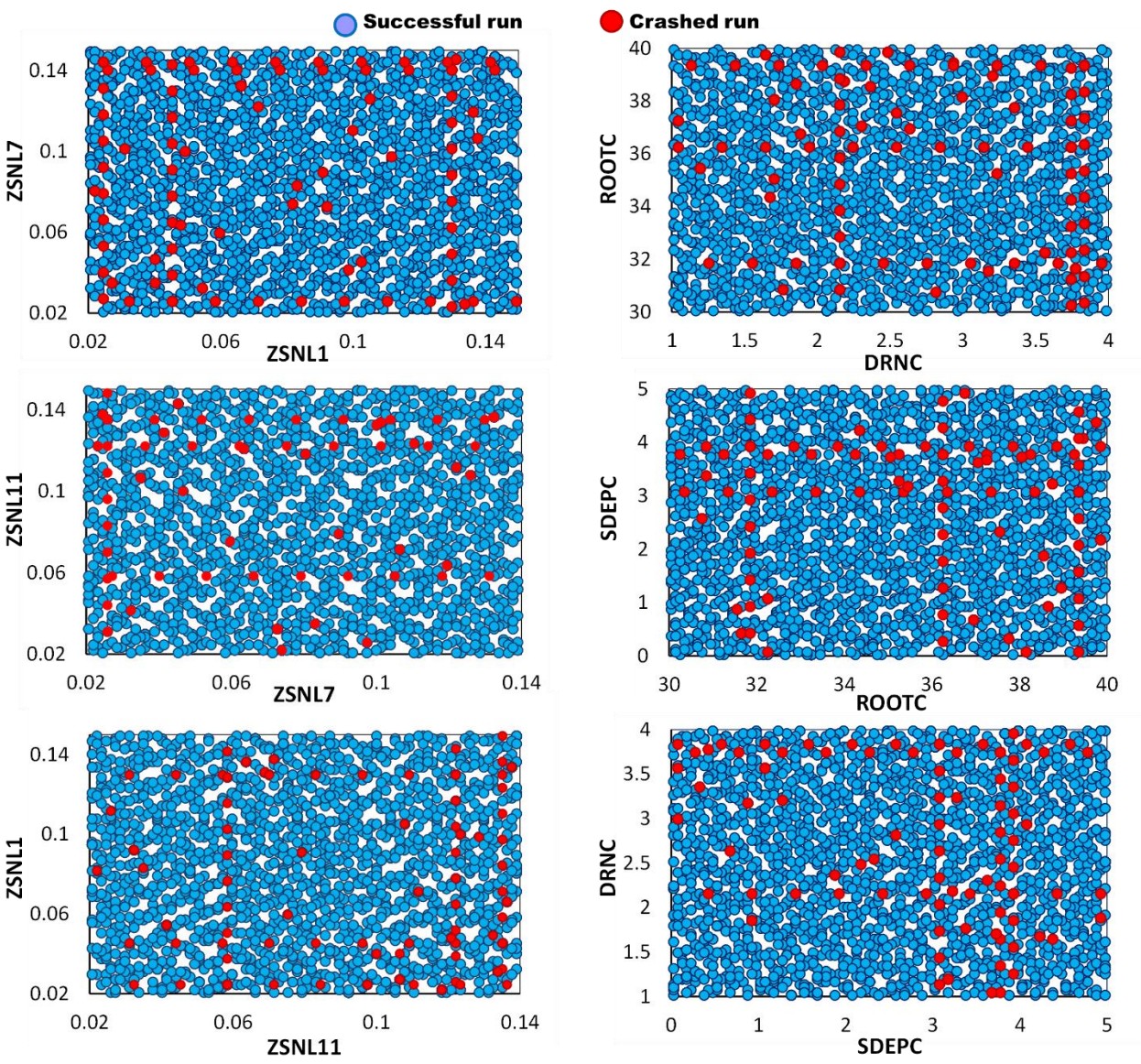

**Figure 15: A 2-D projections of the MESH parameters for successful (blue dots) and crashed (red dots) simulations. Left column shows the threshold snow depth parameters *ZSNL* and right columns shows soil permeable depth (*SDEP*), maximum rooting depth (*ROOT*), and drainage index (*DRN*) for crop vegetation type (C).**

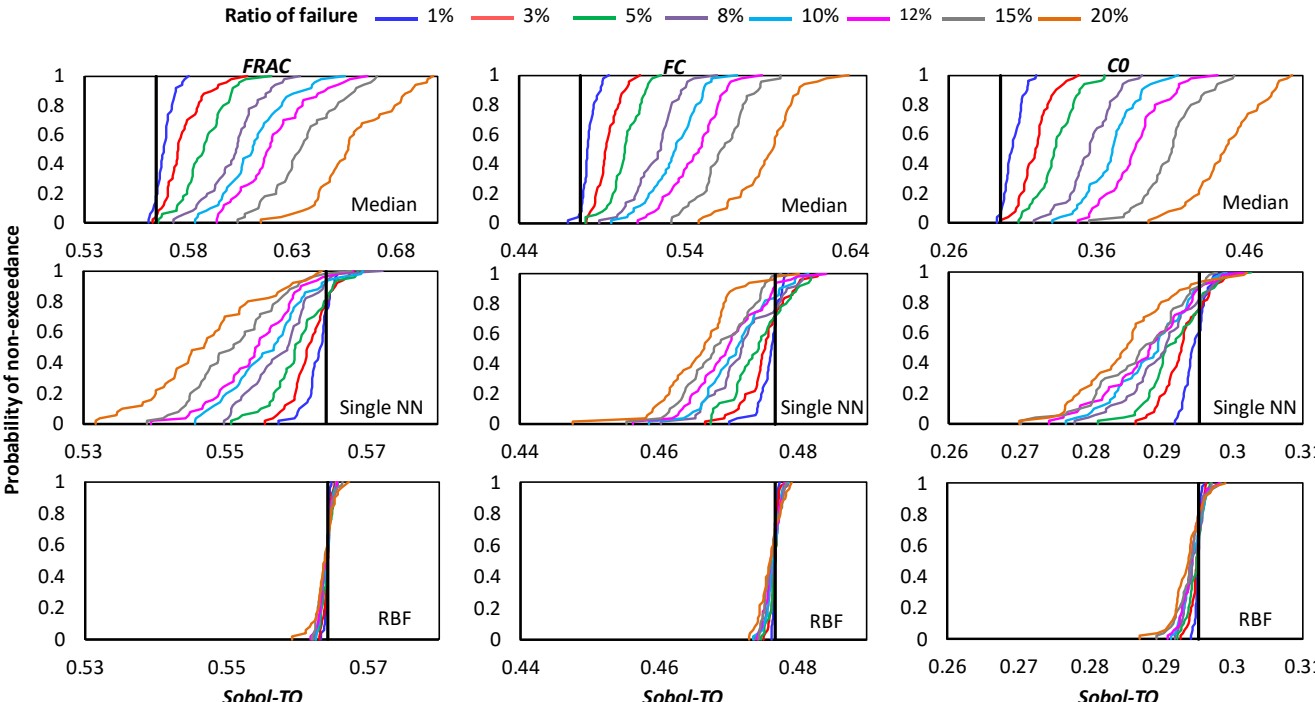

Figure B1: Comparison of the proposed crash handling strategies in sensitivity analysis of the HBV-SASK model using the variance-based algorithm for different ratios of failures. The CDFs of the sensitivity indices for strongly influential parameters {*FRAC*, *FC*, *C0*} are compared in this plot. The vertical line (solid black) on each subplot represents the corresponding ''true'' sensitivity index obtained when there were no failures.

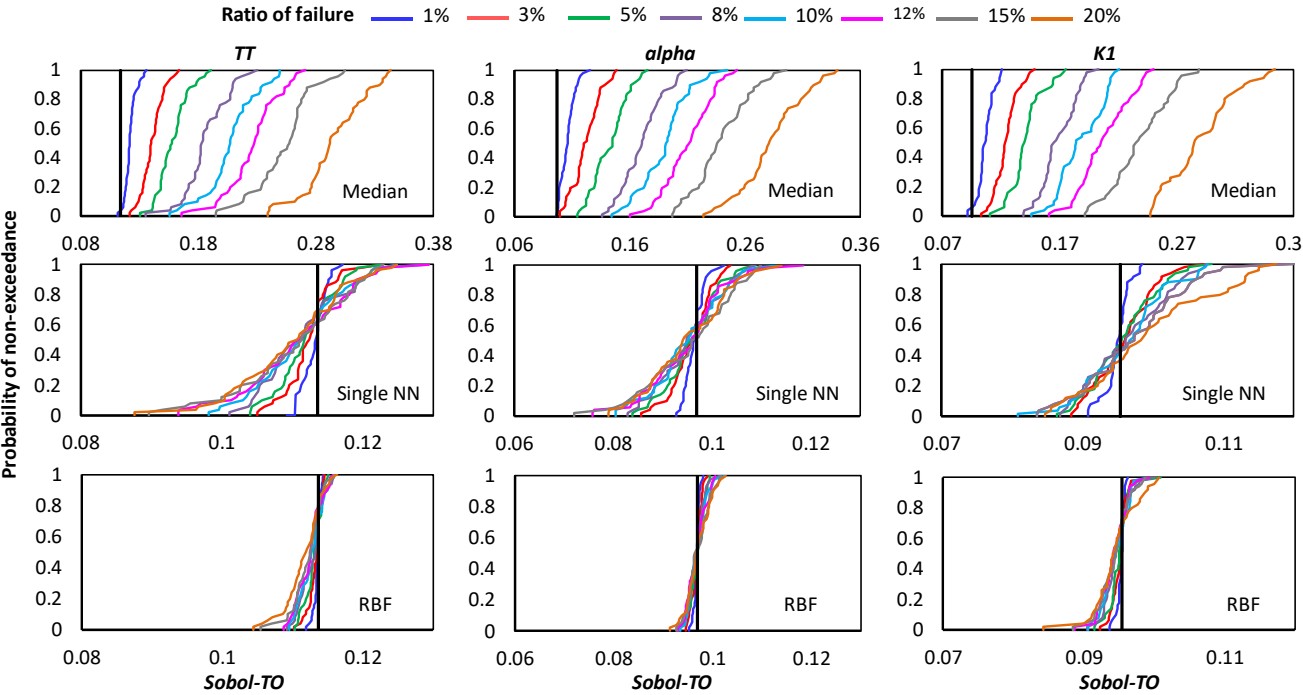

5      **Figure B2: Comparison of the proposed crash handling strategies in sensitivity analysis of the HBV-SASK model using the variance-based algorithm for different ratios of failures. The CDFs of the sensitivity indices for (b) moderately influential parameters {*TT*, *alpha*, *K1*} are compared in this plot. The vertical line (solid black) on each subplot represents the corresponding "true" sensitivity index obtained when there were no failures.**

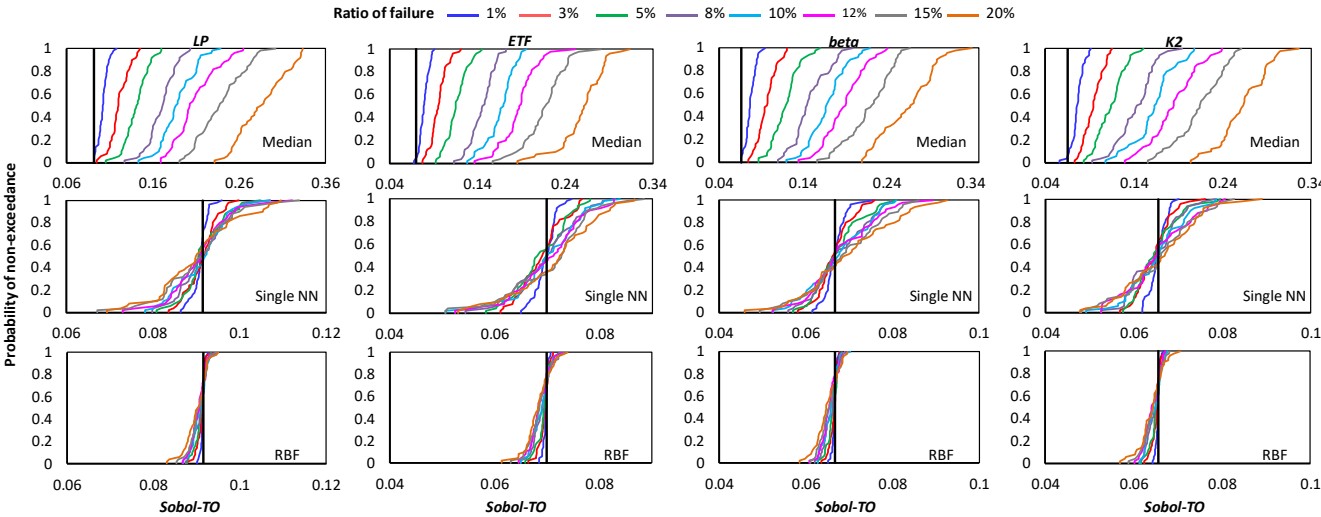

5 **Figure B3: Comparison of the proposed crash handling strategies in sensitivity analysis of the HBV-SASK model using the variance-based algorithm for different ratios of failures. The CDFs of the sensitivity indices for weakly influential parameters (*LP*, *ETF*, *beta*, *K2*) are shown in this plot. The vertical line (solid black) on each subplot represents the corresponding ''true'' sensitivity index obtained when there were no failures.**

**Table 1. HBV-SASK model parameters and their feasible ranges, used in this study. For information on the full parameter set, refer to Razavi et al. (2019).**

| Parameter | Range | Description |
|-----------|-------|-------------|
| *TT* | [-4,4] | Air temperature threshold in °C for melting/freezing and separating rain and snow |
| *C0* | [0,10] | Base melt factor, in mm/°C per day |
| *ETF* | [0,1] | Temperature anomaly correction in 1/°C of potential evapotranspiration |
| *LP* | [0,1] | Limit for PET as a multiplier to FC, i.e., soil moisture below which evaporation becomes supply limited |
| *FC* | [50,500] | Field capacity of soil, in mm. The maximum amount of water that the soil can retain |
| *beta* | [1,3] | Shape parameter (exponent) for soil release equation (unitless) |
| *FRAC* | [0.1,0.9] | Fraction of soil release entering fast reservoir (unitless) |
| *K1* | [0.05,1] | Fast reservoir coefficient, which determines what proportion of the storage is released per day (unitless) |
| *alpha* | [1,3] | Shape parameter (exponent) for fast reservoir equation (unitless) |
| *K2* | [0,0.05] | Slow reservoir coefficient which determines what proportion of the storage is released per day (unitless) |

**Table A1. Grouping of 111 MESH model parameters. These groups are numbered in order of importance.**

| Group number | Parameters |
|---|---|
| 1 | *SDEPC, WFR22, ZSNL3, DRNC, VPDAC, ZPLS4, SDEPD, ROOTC, SDEPG, XSLPC, RATIOs, ZSNL4, ZSNL1, ZPLG4, DDENC, VPDAD, LAMIND, VPDAG, LNZ0D* |
| 2 | *CLAYSa3, SANDSa2, LAMAXC, XSLPD, SANDSa1, RSMNC, ROOTG, ZSNL11, ZSNL7, XSLPG, ZPLG3, ZPLS3, ZPLS1, ZPLG1, DDEND, CLAYSi3, SANDSi3, LNZ0G, SANDSa3, CLAYSa2, CLAYSa1, QA50C, DRNG, VPDBC, DRND, DDENG, LAMAXG, THLQ3, CLAYSi1, SANDSi2, SANDCL3, QA50D, GRKFC, LNZ0C, ALICC, ALVCC, CLAYSi2, ALICG, SANDCL2, SANDCL1, TBAR2, PSGAC, THLQ1, ORGSi3, ORGSi1, PSGBC, THLQ2, TBAR3, TPOND, TBAR1, CMASC, MANNC, ZPOND, RATIOSi, QA50G, RSMNG, RSMND, ORGSi2, RATIOCL, CLAYCL3, GRKFD, CMASD, ORGSa3, ORGSa2, ORGSa1, ORGCL1, ORGCL2, CLAYCL2, ORGCL3, CLAYCL1, ALICD, LAMAXD, ALVCG, GRKFG, ALVCD, VPDBG, CMASG* |
| 3 | *ZPLS11, VPDBD, ZPLG11, PSGAG, PSGBG, LAMING, PSGAD, PSGBD, MANNG, ROOTD, ZPLG7, ZPLS7, TCANO, MANND* |

