# Peer review of "What should we do when a model crashes? Recommendations for global sensitivity analysis of earth and environmental systems models"

_Geoscientific Model Development, 2019_

## Referee Comment (RC1) · Anonymous Referee #1 · 12 Mar 2019

With great interest I have read and reviewed the manuscript "What do we do with model simulation crashes? Recommendations for global sensitivity analysis of earth and environmental systems models" by Sheikholeslami et al. In general, the paper presents a novel and interesting approach to deal with the issue of model crashes when applying global sensitivity analysis. Although this new idea looks promising, additional investigations and explanations are necessary before this paper can be published. In the next sections general, major and minor comments and suggestions are provided that should allow the authors to improve their manuscript.

[Figure]

General To improve the validity of this novel idea and allow it to be applied in a more general context, more investigation is required by

* Applying this for different SA techniques (e.g. a variance based techniques, as one of the proposed approaches might influence the variance of the output) (in particular on the HBV example with different ratios of the number of crashes)

* Apply the k-NN technique instead or next to the simple NN, as the former seems to be more powerful

* Apply a convergence analysis on the SA results. It appears to me that the proposed approach only slows down the convergence of the results (so an evolution of the SA-statistics for both the simulations without crashes and the simulations with suggested crashes should be performed. Possible approaches can be found in (Sarrazin et al., 2016) or (Nossent et al., 2011))

* Different sampling techniques (as the density of the samples might have an influence on the results) (e.g. on p13, L33 one could argue that this statement should be supported by applying different strategies next to "STAR")

* Adding information on the computation time of the different steps

Major

* p2, L27: In most cases, the samples for GSA are independent. This is important in the interpretation of your proposed strategy, so you should clearly mention this in the text.

* p4, L22: In many cases of GSA, it is not necessary to re-run the entire experiment, but just a limited number of runs. This is important to put this into perspective.

* p10, L11: What about parameter "CO"? It is influential, but you don't talk about that one.

* p12, L18: I have the impression that you went into detail too much on these causes of

crashes of this model, whereas your main focus should be on the SA. The part starting on p13, L10 can be maintained as it is an interesting addition to this topic.

* p13, L26: Is this valid for both single NN and k-NN? Specify this.

* p25, fig 7: Could you provide some additional figures of this type in annex? Although this is arbitrarily chosen, this would support the results.

Minor

* p1, L28: Should it be "Dynamical Earth Systems Models" or "Dynamical Earth System Models"?

* p2, L29: Remove either "that" or "how"

* p3, L21: Replace "is" by "are"

* p3, L24: Add "the" before "parameter space"

* p4, L24: Replace "the" by "a"

* p5, L21: It would be either "a computationally simple method" or "computationally simplest method"

* p6, L18: Add "a" before "response"

* p6, L20: Add a comma after "In the literature"

* p6, L22: Add "the" before "RBF"

* p8, L7: Add "a" before "highly"

* p8, L8: Add "a" before "minimum"

* p8, L15: Add "a" or "the" before "maximum"

* p8, L16: Add "the" before "output"

* p8, L16: Add "a" before "minimum"

* p8, L24 (and others): All superscript numbers seem to be written as normal numbers

* p8, L26: "of which" should go before "10"

* p8, L29: The last sentence seems to have an odd structure

* p9, L20: Add "the" before "STAR-VARS

* p9, L25: Add "the" before "GSA"

* p9: L30: I would suggest to move "when there are no crashes" between the brackets on the previous line ("after 9100 function evaluations").

* p10, L7: Add "the" before "parameter space"

* p10, L7: Remove the "s" from "ratios"

* p10, L10: Reformulate this sentence

* p11, L1: Add "the" before "Four" and before "water"

* p11, L10: Replace "with" by "between these"

* p11, L12: Add "a" before "vegetation"

* p11, L13: Add "the" before "soil"

* p11, L21: Reformulate "As shown"

* p11, L23: Add an "s" to "order"

* p11, L29: Remove the "7"

* p13, L30: Replace "depends" by "depending"

* p14, L1: Which feature? Reformulate this sentence

* p14, L6: Replace "are" by "should be"

* p14, L7: Add an "s" to "problem"

* p14, L25: Remove the "s" from "involves"

* p14, L31: "The efficiency of our proposed simulation based strategies was shown…"

* p14, L30: This is a very long, complex sentence.

* p15, L11: "causing" instead of "casing"

* p15, L18: "understanding" instead of "understand"

* p23 caption: Remove "C0" from the list of "moderately influential parameters"

Ref

Nossent, Elsen and Bauwens. Sobol' sensitivity analysis of a complex environmental model, Environmental Modelling & Software, 26 (12), 2011

Sarrazin, Pianosi and Wagener. Global Sensitivity Analysis of environmental models: convergence and validation. Environmental Modelling & Software, 79, 2016

---

## Referee Comment (RC2) · Anonymous Referee #2 · 28 Jun 2019

The authors argue to substitute data of failed simulation members in large ensemble simulations conducted for global parametric sensitivity analysis of dynamical earth system models. It is common for the models to crash for certain parameter value combinations that are randomly sampled from multidimensional parameter space using standard automated techniques. Using case studies, the authors show that it may be better to fill in the data from the failed experiments with data substitution techniques rather than the general practice of ignoring those experiments completely. The paper is generally well written and motivated. I point out my concerns below.

[Figure]

1. The authors motivate the study well (Section 1.2). However, the authors state that the automated sampling method that they use - STAR-VARS breaks down if there are failed simulations for certain parameter combinations (Section 2.3). They do not provide a good reasoning for that, which I think is warranted. Are there other sampling methods that would not be sensitive to failed simulations? Why use STAR-VARS? Is the data substitution strategy only designed because of the limitation of STAR-VARS?

2. The impact of three data substitution techniques are compared. However, the first two methods are overly simplistic, and one can argue that they would yield poorer results a-priori - for example, the median is definitely not a good approximation for parameter combinations that are in the distribution tails, which may be more likely to crash. I do not see why the authors chose to present the results from those methods as one of their main results. It is fine to include them, but I think it would have been more useful to include results from different surrogate models, e.g. krigging, neural networks etc., which may be better as models for data substitution.

3. The authors appear to consider simulation failure as numerical artefacts. It could well be that parameter combinations are unphysical resulting in genuine crashes. Substituting data for these model crashes would result in unrealistic sensitivity. Likewise, unrealistic parameter combinations could also result in successful runs without crashes distorting the sensitivity analysis. It will be good if the authors could discuss this. The authors discuss this partly in section 5.1 for MESH model while exploring the reasons of simulation failure, but do not seem to relate it to their substitution strategy which is their main point.

4. The title reads as if something useful can be done with simulations that crashed. But, the strategy of the paper is to actually substitute the failed simulations. The authors should think about revising the title so that its not too misleading.

---

## Author Comment (AC1) · 1 Aug 2019

This document contains copies of all comments of the Reviewer 1 and our planned efforts to address them in the revised manuscript.

Reviewer 1: With great interest I have read and reviewed the manuscript "What do we do with model simulation crashes? Recommendations for global sensitivity analysis of earth and environmental systems models" by Sheikholeslami et al. In general, the paper presents a novel and interesting approach to deal with the issue of model
crashes when applying global sensitivity analysis. Although this new idea looks promising, additional investigations and explanations are necessary before this paper can be published. In the next sections general, major and minor comments and suggestions are provided that should allow the authors to improve their manuscript.

Response: We are very thankful to the reviewer for the time and effort spent on reviewing our paper. The comments and suggestions were constructive and helped us improve the quality of our manuscript. We will address all the comments in the revised manuscript as described in this rebuttal document.

Reviewer 1: General. To improve the validity of this novel idea and allow it to be applied in a more general context, more investigation is required by:

- Applying this for different SA techniques (e.g. a variance-based technique, as one of the proposed approaches might influence the variance of the output) (in particular on the HBV example with different ratios of the number of crashes).

Response: Thanks for this important comment. As suggested by the reviewer, we have tested the proposed crash handling strategy using a variance based GSA technique and compared the results for the first case study. We will add these results into the revised manuscript. However, note that our proposed strategy is GSA method-free and does not depend on the utilized GSA algorithm.

- Applying the k-NN technique instead or next to the simple NN, as the former seems to be more powerful.

Response: We agree that the k-NN technique is an effective regression approach compared to the single NN. Considering our main goal in this study, i.e., finding simple yet effective strategies for handling simulation failures, we have adopted the single NN technique because it is a simple, persimmons method. Of course, more complex metamodeling options are available, including k-NN. The objective was not to provide a comprehensive comparison though. We have added the following paragraph in Section

2.2.2 to address this comment:

"The single NN is a parsimonious, very simple to understand, and equally easy to implement. To fill-in the crashed simulations, single-NN algorithm reads through whole dataset to find the nearest neighbor. On the other hand, the k-NN technique is more complex and requires a careful selection of the kernel functions to assign weights based on the degree of similarity between nearest neighbors. The choice of kernel functions and variable k is subjective. Particularly, k is a hyperparameter that user must pick to get the best possible fit for the dataset. However, the optimal k will always vary depending on the dataset. It is noteworthy that some authors have asserted that covariances among Y variables are preserved in the NN-based techniques when using small k-values (Hudak et al., 2008; LeMay & Temesgen, 2005; McRoberts et al., 2002; Tomppo et al., 2002). McRoberts (2009) showed that the variance and covariance of the Y variables tend to be preserved for k = 1 but not for k > 1 (McRoberts, 2009)."

- Applying a convergence analysis on the SA results. It appears to me that the proposed approach only slows down the convergence of the results (so an evolution of the SA statistics for both the simulations without crashes and the simulations with suggested crashes should be performed. Possible approaches can be found in (Sarrazin et al., 2016) or (Nossent et al., 2011)).

Response: We think there is no point in this comparison, as when a model crashes the classic convergence analysis does not make sense because the GSA algorithm cannot be finished. In other words, since crash strategies are applied after all runs are completed and without any need to repeat any experiment, crash strategies should not matter for convergence. To emphasize on the importance of the convergence analysis for the GSA, the following paragraph has been added into the discussion section (Section 5.2):

"It is important to note that the sample size in GSA studies should not only be determined based on the available computational budget but also considerations of GSA

stability and convergence. Therefore, it is of vital importance to monitor and evaluate the convergence rate of the GSA algorithms. Strategies introduced by Nossent et al. (2011), Sarrazin et al., 2016, and more recently by Sheikholeslami et al. (2019) enable users to diagnose the convergence behavior of the GSA algorithms."

- Applying different sampling techniques (as the density of the samples might have an influence on the results) (e.g. on p13, L33 one could argue that this statement should be supported by applying different strategies next to "STAR").

Response: Thanks for pointing out this comment. As mentioned in Section 5.2, regardless of the chosen method for handling simulation crash problem in GSA, it is advisable to spend some time up front to find an optimal sample set before submitting it for evaluation to the computationally expensive models. Therefore, we used an advanced sampling strategy called Progressive Latin Hypercube Sampling (PLHS) to ensure sufficient coverage of the parameter space (please see Section 2.3). In other words, the STAR-VARS algorithm employs the PLHS strategy to locate star centers in the first phase of sampling.

- Adding information on the computation time of the different step.

Response: To address this comment, the following sentences have been added to the revised manuscript:

"The entire set of 100,000 function evaluations of the MESH model will take more than 6 months if a single CPU core is used. However, we used the University of Saskatchewan's high-performance computing system to run the GSA experiment in parallel on 160 cores. Therefore, completing all model runs required approximately 32 hours."

"For the MESH model, using an Intel$^®$ Core$^{TM}$ i7 CPU 4790 3.6GHz desktop PC, the RBF technique took only 65 seconds to substitute 3,084 crashed runs, while the single NN technique required about 197 seconds to complete the task."

Reviewer 1: Major comments:

* p2, L27: In most cases, the samples for GSA are independent. This is important in the interpretation of your proposed strategy, so you should clearly mention this in the text.

Response: Thanks to Reviewer 1 for this important comment. As we mentioned in the introduction section, in case of model crashes, re-running the entire experiment is inevitable when using a GSA technique that utilizes a sampling strategy with a particular structure. To address this comment, we have revised Section 1.2 as follows:

"Ignoring the crashed runs in GSA is only relevant when using purely random (and independent) samples (i.e., Monte Carlo method). In such cases, if the model crashes at a given parameter set, one can simply exclude that parameter set or repeat randomly generating a parameter set (at the expense of increased computational cost) that results in a successful simulation."

"Reducing the number of model runs and finding optimum locations for sample points in the parameter space are the main drivers for implementing improved sampling techniques in GSA. Typically, these sampling techniques are not only random but also follow specific spatial arrangements. However, when applying a sampling-based technique that uses an ad-hoc sampling strategy with particular spatial structure (e.g., the variance-based GSA proposed by Saltelli et al. (2010) or STAR-VARS of Razavi and Gupta (2016b)), we cannot ignore crashed simulations. In this case, excluding sample points associated with simulation crashes will distort the structure of the sample set, causing inaccurate estimation of sensitivity indices. As a result, the user may have to re-do a part or the entire experiment depending on the GSA implementation, by generating a new sample set (or a succession of sample sets), leading to a waste of previous model runs."

* p4, L22: In many cases of GSA, it is not necessary to re-run the entire experiment, but just a limited number of runs. This is important to put this into perspective.

Response: Agreed. Please see our response to the previous comment.

* p10, L11: What about parameter "CO"? It is influential, but you don't talk about that one.

Response: Thanks for pointing this out. To address this comment, we have edited the text as follows:

"From the results, we see that using the RBF technique the sensitivity indices of the most important parameters (FRAC, FC, C0) (Fig. 4(a)) and less important parameters (LP, ETF, beta, K2) (Fig. 5) were estimated with high degree of accuracy and robustness."

* p12, L18: I have the impression that you went into detail too much on these causes of crashes of this model, whereas your main focus should be on the SA. The part starting on p13, L10 can be maintained as it is an interesting addition to this topic.

Response: As suggested by the reviewer, the length of this section has been reduced in the revised manuscript.

* p13, L26: Is this valid for both single NN and k-NN? Specify this.

Response: Yes, the poorly sampled parameter space can also influence the performance of the k-NN technique. In regions of the parameter space where the sample points are sparsely distributed, distances to nearest neighbours can be high, leading to choosing physically incompatible neighbours. To clarify this point, we have modified the text as follows:

"For example, in the NN techniques (both single and k-NN) one major concern is that the sparseness of sample points may affect the quality of the results."

* p25, fig 7: Could you provide some additional figures of this type in annex? Although this is arbitrarily chosen, this would support the results.

Response: We agree with the reviewer that reporting these figures might be of some

value but adding these results would not change the discussions and conclusions already presented, and so we prefer not to add more figures to our already long paper having many figures. To further investigate the performance of the single NN and RBF techniques, we have added a new figure (the bottom panel of the Fig.7) into the revised manuscript.

Reviewer 1: Minor comments:

Response: Thank you very much for editing and proofreading our manuscript.

* p1, L28: Should it be "Dynamical Earth Systems Models" or "Dynamical Earth System Models"?

Response: As suggested, we used "Dynamical Earth System Models".

* p2, L29: Remove either "that" or "how"

Response: Fixed.

* p3, L21: Replace "is" by "are"

Response: Typo fixed.

* p3, L24: Add "the" before "parameter space"

Response: Fixed.

* p4, L24: Replace "the" by "a"

Response: Fixed.

* p5, L21: It would be either "a computationally simple method" or "computationally simplest method"

Response: Corrected.

* p6, L18: Add "a" before "response"

[Figure]

Response: Fixed.

* p6, L20: Add a comma after "In the literature"

Response: Fixed.

* p6, L22: Add "the" before "RBF"

Response: Fixed.

* p8, L7: Add "a" before "highly"

Response: Fixed.

* p8, L8: Add "a" before "minimum"

Response: Fixed.

* p8, L15: Add "a" or "the" before "maximum"

Response: Fixed.

* p8, L16: Add "the" before "output"

Response: Fixed.

* p8, L16: Add "a" before "minimum"

Response: Fixed.

* p8, L24 (and others): All superscript numbers seem to be written as normal numbers

Response: Typo fixed.

* p8, L26: "of which" should go before "10"

Response: Corrected as suggested.

* p8, L29: The last sentence seems to have an odd structure

Response: It has been modified.

* p9, L20: Add "the" before "STAR-VARS"

Response: Fixed.

* p9, L25: Add "the" before "GSA"

Response: Fixed.

* p9: L30: I would suggest to move "when there are no crashes" between the brackets on the previous line ("after 9100 function evaluations").

Response: Corrected as suggested.

* p10, L7: Add "the" before "parameter space"

Response: Fixed.

* p10, L7: Remove the "s" from "ratios"

Response: Typo fixed.

* p10, L10: Reformulate this sentence

Response: We have modified p10, L10 and now it reads:

"Moreover, Fig. 5 shows that when crashes were substituted using the RBF technique, the STAR-VARS algorithm estimated the sensitivity indices of the most important parameters (FRAC, FC, C0) (Fig. 4(a)) and less important parameters (LP, ETF, beta, K2) (Fig. 5) with higher degrees of accuracy and robustness."

* p11, L1: Add "the" before "Four" and before "water"

Response: Fixed.

* p11, L10: Replace "with" by "between these"

Response: Corrected.

* p11, L12: Add "a" before "vegetation"
Response: Fixed.

* p11, L13: Add "the" before "soil"

Response: Fixed.

* p11, L21: Reformulate "As shown"

Response: It has been edited as follows:

"The STAR-VARS algorithm identified these parameters as weakly influential (very low IVARS-50 values) using the proposed crash handling techniques. However, the associated sensitivity indices obtained by the RBF imputation method are two orders of magnitude larger for the parameters in the left panel (Fig.11 (a, c)) and about four orders of magnitude larger for the parameters in the right panel (Fig. 11 (b, d)) compared to those obtained by the single NN and median substitution methods."

* p11, L23: Add an "s" to "order"

Response: Fixed.

* p11, L29: Remove the "7"

Response: Fixed.

* p13, L30: Replace "depends" by "depending"

Response: Typo fixed.

* p14, L1: Which feature? Reformulate this sentence

Response: Corrected.

* p14, L6: Replace "are" by "should be"

Response: Corrected.

* p14, L7: Add an "s" to "problem"

Response: Typo fixed.

* p14, L25: Remove the "s" from "involves"

Response: Typo fixed.

* p14, L31: "The efficiency of our proposed simulation based strategies was shown: : :"

Response: Fixed.

* p14, L30: This is a very long, complex sentence.

Response: This sentence has been reformulated in the revised manuscript and now it reads:

"The efficiency of our proposed substitution-based strategy was shown to be remarkable compared to other crash handling strategies (ignorance-based and non-substitution procedures). This is of prominent importance particularly when dealing with GSA of the computationally expensive models mainly because the proposed strategy does not need repeating the entire experiment."

* p15, L11: "causing" instead of "casing"

Response: Typo fixed.

* p15, L18: "understanding" instead of "understand"

Response: Typo fixed.

* p23 caption: Remove "C0" from the list of "moderately influential parameters"

Response: Corrected.

---

## Author Comment (AC2) · 2 Aug 2019

This document contains copies of all comments of the Reviewer 2 and our planned efforts to address them.

Reviewer 2: The authors argue to substitute data of failed simulation members in large ensemble simulations conducted for global parametric sensitivity analysis of dynamical earth system models. It is common for the models to crash for certain parameter value combinations that are randomly sampled from multidimensional parameter space using

standard automated techniques. Using case studies, the authors show that it may be better to fill in the data from the failed experiments with data substitution techniques rather than the general practice of ignoring those experiments completely. The paper is generally well written and motivated. I point out my concerns below.

Response: We greatly appreciate Reviewer 2 for reviewing the manuscript and providing positive evaluations.

1. The authors motivate the study well (Section 1.2). However, the authors state that the automated sampling method that they use - STAR-VARS breaks down if there are failed simulations for certain parameter combinations (Section 2.3). They do not provide a good reasoning for that, which I think is warranted. Are there other sampling methods that would not be sensitive to failed simulations? Why use STAR-VARS? Is the data substitution strategy only designed because of the limitation of STAR-VARS?

Response to 1: Thanks for this comment. As we mentioned in Section 1.2, those GSA techniques that use a sampling strategy with a specific structure will fail if the simulation model crashes at certain parameter configurations such as the widely-used variance-based method of Saltelli et al. (2010). To further explain, we have revised Section 1.2 as follows:

"Ignoring the crashed runs in GSA is only relevant when using purely random (and independent) samples (i.e., Monte Carlo method). In such cases, if the model crashes at a given parameter set, one can simply exclude that parameter set or repeat randomly generating a parameter set (at the expense of increased computational cost) that results in a successful simulation."

"Reducing the number of model runs and finding optimum locations for sample points in the parameter space are the main drivers for implementing improved sampling techniques in GSA. Typically, these sampling techniques are not only random but also follow specific spatial arrangements. However, when applying a sampling-based technique that uses an ad-hoc sampling strategy with particular spatial structure (e.g., the

variance-based GSA proposed by Saltelli et al. (2010) or STAR-VARS of Razavi and Gupta (2016b)), we cannot ignore crashed simulations. In this case, excluding sample points associated with simulation crashes will distort the structure of the sample set, causing inaccurate estimation of sensitivity indices. As a result, the user may have to re-do a part or the entire experiment depending on the GSA implementation, by generating a new sample set (or a succession of sample sets), leading to a waste of previous model runs."

2. The impact of three data substitution techniques are compared. However, the first two methods are overly simplistic, and one can argue that they would yield poorer results a-priori - for example, the median is definitely not a good approximation for parameter combinations that are in the distribution tails, which may be more likely to crash. I do not see why the authors chose to present the results from those methods as one of their main results. It is fine to include them, but I think it would have been more useful to include results from different surrogate models, e.g. kriging, neural networks etc., which may be better as models for data substitution.

Response to 2: We certainly agree with reviewer that the median substitution is a very simple (perhaps naive) approach compared to other methods such as RBF. Nevertheless, we have adopted these methods considering our main goal in this study, which was finding simple and effective strategies for handling simulation failures. To improve the explanation, we have added the following statement to Section 2.2.1:

"In sampling-based optimization, assigning a very large objective function value for the parameter configurations violating the problem constraints is a common practice (known as the big M method). Inspired by the big M method we used the median substitution technique in this paper. However, since replacing crashes with a big value can magnify the effect of the crashed runs in GSA, it is reasonable to choose a central value such as median to minimize the impact of the implausible parameter configurations."

Regarding the application of other surrogate models (e.g., kriging, etc.) as we mentioned in Section 2.2.3 depending on the complexity and dimensionality of the response surface, other types of metamodels can be incorporated into the proposed framework. We did not intend to compare the performance of different metamodelling techniques in this study, so we only applied the well-known RBF technique. Of course, this could be a potential direction for future research. To address this comment, we have added the following sentence to the conclusion section of the revised manuscript:

"In addition, future work may include application and testing of other types of emulation techniques such as kriging and support vector machine to handle simulation failures."

3. The authors appear to consider simulation failure as numerical artefacts. It could well be that parameter combinations are unphysical resulting in genuine crashes. Substituting data for these model crashes would result in unrealistic sensitivity. Likewise, unrealistic parameter combinations could also result in successful runs without crashes distorting the sensitivity analysis. It will be good if the authors could discuss this. The authors discuss this partly in section 5.1 for MESH model while exploring the reasons of simulation failure, but do not seem to relate it to their substitution strategy which is their main point.

Response to 3: Thanks for this very good comment. The following paragraph has been added to Section 5.1 of the revised manuscript to improve the discussion:

"We conclude this section by highlighting a point that should receive careful attention when applying the substitution-based methods in handling model crashes. In addition to the numerical artifacts in simulation models, some combinations of parameter values, which are not physically valid, can also lead to simulation failures. As a result, substituting data for these crashed runs would result in unrealistic parameter sensitivity. Preventing this type of failure requires knowledge about the reasonable parameter ranges in DESMs. This type of crash can be significantly reduced by selecting plausible ranges of parameters based on physical knowledge or information of the problem (a process referred to as "parameter space refinement" (see e.g., Li et al., 2019;

Williamson et al., 2013)). However, DESMs often consist of many interacting, uncertain parameters, and therefore very little may be known a priori about the implausible regions of the parameter space. "

4. The title reads as if something useful can be done with simulations that crashed. But, the strategy of the paper is to actually substitute the failed simulations. The authors should think about revising the title so that its not too misleading.

Response to 4: We greatly appreciate the reviewer for this valuable suggestion. The new title now reads:

"What should we do when a model crashes? Recommendations for global sensitivity analysis of earth and environmental systems models"

---

## Author Response (AR1)

Dear Dr. Easterbrook and Reviewers:

Thank you for your initial consideration of our manuscript (gmd-2019-17, "What do we do with model simulation crashes? Recommendations for global sensitivity analysis of earth and environmental systems models"). We appreciate the detailed suggestions for improvement and opportunity to submit a revised version. In response to the reviewers' comments, we have made major revisions to the manuscript and addressed all the comments as described in this rebuttal document. These are detailed in a point-by-point response to each comment below; reviewer comments are in *italicized*, *blue text* and our response is in normal, black text.

The reviewers' comments have substantially improved the manuscript in a number of ways. Major changes have been made to the revised manuscript, as listed below:

1) The title of paper has been modified in the revised manuscript as suggested by Reviewer 2.
2) In response to the comments raised by Reviewer 1, another sensitivity analysis method (i.e., a variance-based method) has been applied in order to demonstrate the utility of the proposed crash handling approach.
3) A new figure has been added into the results section to further compare the RBF and single NN techniques in terms of approximating the response surface.
4) New papers have been cited in the revised manuscript to demonstrate the motivation of our work and its merit over previous studies.

Other major/minor modifications have also been made to the manuscript to address the review comments, which have been listed below their associated review comments in the following. All these revisions have enhanced the contribution of our study to the literature by providing an efficient and effective approach to cope with failed simulations when performing global sensitivity analysis. We believe this is a valuable contribution to the Earth and Environmental Systems Modelling community and will be of great interest to Geoscientific Model Development readers.

Thank you for your consideration, and we hope to hear from you soon,

Razi Sheikholeslami et al.
razi.sheikholeslami@usask.ca

**Response to Reviewers' Comments**

This document contains copies of all the comments of Editor and Reviewers (in *italicized, blue text*) and our subsequent efforts to address them (in normal, black text).

**Reviewer 1**

*With great interest I have read and reviewed the manuscript "What do we do with model simulation crashes? Recommendations for global sensitivity analysis of earth and environmental systems models" by Sheikholeslami et al. In general, the paper presents a novel and interesting approach to deal with the issue of model crashes when applying global sensitivity analysis. Although this new idea looks promising, additional investigations and explanations are necessary before this paper can be published. In the next sections general, major and minor comments and suggestions are provided that should allow the authors to improve their manuscript.*

**Response**: We are very thankful to the reviewer for the time and effort spent on reviewing our paper. The comments and suggestions were constructive and helped us improve the quality of our manuscript.

**General**

*To improve the validity of this novel idea and allow it to be applied in a more general context, more investigation is required by:*

*- Applying this for different SA techniques (e.g. a variance-based technique, as one of the proposed approaches might influence the variance of the output) (in particular on the HBV example with different ratios of the number of crashes).*

**Response**: Thanks for this important comment. As suggested by the reviewer, we have tested the proposed crash handling approach using a variance based GSA technique and compared the results for the first case study in the revised manuscript (see **Figure 8** and **Appendix B**). However, note that since our proposed strategy is GSA-method-free and does not depend on the utilized GSA algorithm, the results have not changed, i.e., the RBF and single NN techniques still outperformed the median substitution in terms of closeness to the true GSA results and robustness when crashes happened at different locations of the parameter space.

*- Applying the k-NN technique instead or next to the simple NN, as the former seems to be more powerful.*

**Response**: We agree that the $k$-NN technique is an effective regression approach compared to the single NN. Considering our main goal in this study, i.e., finding simple yet effective strategies for handling simulation failures, we have adopted the single NN technique because it is a simple, persimmons method. Of course, more complex metamodeling options are available, including $k$-NN. The objective was not to provide a comprehensive comparison though. We have added the following paragraph in **Section 2.2.2** to address this comment:

> In this study, we choose to use the single NN technique with Euclidean distance measure. We do so because the single NN technique is very parsimonious and simple to understand and implement. To substitute the crashed simulations, the single-NN algorithm reads through whole dataset to find the nearest neighbour and then imputes the missing value with the model response of that nearest neighbour. It is noteworthy that some authors have asserted that covariances among Y variables are preserved in the NN-based techniques when using small k values (Hudak et al., 2008; McRoberts et al., 2002; Tomppo et al., 2002). But, McRoberts (2009) showed that the variance and covariance of the Y variables tend to be preserved for k = 1 but not for k > 1 (McRoberts, 2009). In general,

compared to the single NN-technique, the k-NN technique may provide a better fit to data but at the expense of being more complex and requiring a careful (and subjective) selection of the kernel functions and variable k. As a more complex technique, we suggest directly using a model emulation technique as described in the section below.

*- Applying a convergence analysis on the SA results. It appears to me that the proposed approach only slows down the convergence of the results (so an evolution of the SA statistics for both the simulations without crashes and the simulations with suggested crashes should be performed. Possible approaches can be found in (Sarrazin et al., 2016) or (Nossent et al., 2011)).*

**Response**: Thanks for providing these useful references. We think there is no point in this comparison, as when a model crashes the classic convergence analysis does not make sense because the GSA algorithm cannot be finished. In other words, since the crash handling strategies are applied after all runs are completed and without any need to repeat any experiment, these strategies should not matter for convergence. To emphasize on the importance of the convergence analysis for the GSA, the following paragraph has been added into the discussion section (**Section 5.2**)

It is important to note that the sample size in GSA studies should not only be determined based on the available computational budget but also considerations of GSA stability and convergence. Therefore, it is of vital importance to monitor and evaluate the convergence rate of the GSA algorithms. Strategies introduced by Nossent et al. (2011), Sarrazin et al. (2016), and more recently by Sheikholeslami et al. (2019) enable users to diagnose the convergence behaviour of the GSA algorithms.

*- Applying different sampling techniques (as the density of the samples might have an influence on the results) (e.g. on p13, L33 one could argue that this statement should be supported by applying different strategies next to "STAR").*

**Response**: Thanks for pointing out this comment. As mentioned in **Section 5.2**, regardless of the chosen method for handling simulation crash problem in GSA, it is advisable to spend some time up front to find an optimal sample set before submitting it for evaluation to the computationally expensive models. Therefore, we used an advanced sampling strategy called Progressive Latin Hypercube Sampling (PLHS) to ensure sufficient coverage of the parameter space (see **Section 3.3**). In other words, the STAR-VARS algorithm employs the PLHS strategy to locate star centers in the first phase of sampling. In addition, for the variance-based GSA, we applied Sobol quasi-random sequences combined with skipping, leaping, and scrambling operations to generate the base sample points.

*- Adding information on the computation time of the different step.*

**Response**: To address this comment, the following sentences have been added to **Section 3.3** of the revised manuscript:

The entire set of 100,000 function evaluations of the MESH model would take more than 6 months if we used a single CPU core. However, we used the University of Saskatchewan's high-performance computing system to run the GSA experiment in parallel on 160 cores. Therefore, completing all model runs required approximately 32 hours. For the MESH model, using an Intel® Core™ i7 CPU 4790 3.6GHz desktop PC, the RBF technique took only 65 seconds to substitute 3,084 crashed runs, while the single NN technique required about 97 seconds to complete the task.

**Major**
*\* p2, L27: In most cases, the samples for GSA are independent. This is important in the interpretation of your proposed strategy, so you should clearly mention this in the text.*

**Response**: Thanks to Reviewer 1 for this important comment. As we mentioned in the introduction section, in case of model crashes, re-running the entire experiment is inevitable when using a GSA technique that utilizes a sampling strategy with a particular structure. To address this comment, we have revised **Section 1.2** as follows:

> 3. Ignoring the crashed runs in GSA may only be seen relevant when using purely random (and independent) samples (i.e., Monte Carlo method). In such cases, if the model crashes at a given parameter set, one may simply exclude that parameter set or generate another random parameter set (at the expense of increased computational cost) that results in a successful simulation.

> 4. Some efficient sampling techniques follow specific spatial arrangements; examples include the variance-based GSA proposed by Saltelli et al. (2010) or STAR-VARS of Razavi and Gupta (2016b). In GSA enabled with such structured sampling techniques, we cannot ignore crashed simulations because excluding sample points associated with simulation crashes will distort the structure of the sample set, causing inaccurate estimation of sensitivity indices. As a result, the user may have to re-do a part or the entire experiment depending on the GSA implementation.

*\* p4, L22: In many cases of GSA, it is not necessary to re-run the entire experiment, but just a limited number of runs. This is important to put this into perspective.*

**Response**: Agreed. Please see our response to previous comment.

*\* p10, L11: What about parameter "CO"? It is influential, but you don't talk about that one.*

**Response**: Thanks for pointing this out. To address this comment, we have edited the text as follows:

> Moreover, Fig. 4 and 6 show that when crashes were substituted using the RBF technique, the STAR-VARS algorithm estimated the sensitivity indices of the most important parameters (FRAC, FC, C0) (Fig. 4) and less important parameters (LP, ETF, beta, K2) (Fig. 6) with high degrees of accuracy and robustness.

*\* p12, L18: I have the impression that you went into detail too much on these causes of crashes of this model, whereas your main focus should be on the SA. The part starting on p13, L10 can be maintained as it is an interesting addition to this topic.*

**Response**: As suggested by the reviewer, the length of this section has been reduced in the revised manuscript.

*\* p13, L26: Is this valid for both single NN and k-NN? Specify this.*

**Response**: Yes, the poorly sampled parameter space can also influence the performance of the $k$-NN technique. In regions of the parameter space where the sample points are sparsely distributed, distances to nearest neighbours can be high, leading to choosing physically incompatible neighbours. To clarify this point, we have modified the text as follows:

> For example, in the NN techniques (both single and $k$-NN) one major concern is that the sparseness of sample points may affect the quality of the results.

*\* p25, fig 7: Could you provide some additional figures of this type in annex? Although this is arbitrarily chosen, this would support the results.*

**Response**: We agree with the reviewer that reporting these figures might be of some value but adding these results would not change the discussions and conclusions already presented, and so we prefer not to add more figures to our already long paper having many figures. To further investigate the performance of the

single NN and RBF techniques, we have added a new figure (the upper panel of the **Figure 9**) into the revised manuscript.

**Minor**

**Response**: Thank you very much for editing and proofreading our manuscript.

*\* p1, L28: Should it be "Dynamical Earth Systems Models" or "Dynamical Earth System Models"?*
**Response**: As suggested, we used "Dynamical Earth System Models".

*\* p2, L29: Remove either "that" or "how"*
**Response**: Fixed.
*\* p3, L21: Replace "is" by "are"*
**Response**: Typo fixed.
*\* p3, L24: Add "the" before "parameter space"*
**Response**: Fixed.
*\* p4, L24: Replace "the" by "a"*
**Response**: Fixed.
*\* p5, L21: It would be either "a computationally simple method" or "computationally simplest method"*
**Response**: Corrected.
*\* p6, L18: Add "a" before "response"*
**Response**: Fixed.
*\* p6, L20: Add a comma after "In the literature"*
**Response**: Fixed.
*\* p6, L22: Add "the" before "RBF"*
**Response**: Fixed.
*\* p8, L7: Add "a" before "highly"*
**Response**: Fixed.
*\* p8, L8: Add "a" before "minimum"*
**Response**: Fixed.
*\* p8, L15: Add "a" or "the" before "maximum"*
**Response**: Fixed.
*\* p8, L16: Add "the" before "output"*
**Response**: Fixed.
*\* p8, L16: Add "a" before "minimum"*
**Response**: Fixed.
*\* p8, L24 (and others): All superscript numbers seem to be written as normal numbers*
**Response**: Typo fixed.
*\* p8, L26: "of which" should go before "10"*
**Response**: Corrected as suggested.
*\* p8, L29: The last sentence seems to have an odd structure*
**Response**: It has been modified.
*\* p9, L20: Add "the" before "STAR-VARS*
**Response**: Fixed.
*\* p9, L25: Add "the" before "GSA"*
**Response**: Fixed.
*\* p9: L30: I would suggest to move "when there are no crashes" between the brackets on the previous line ("after 9100 function evaluations").*
**Response**: Corrected as suggested.
*\* p10, L7: Add "the" before "parameter space"*

**Response**: Fixed.
*\* p10, L7: Remove the "s" from "ratios"*
**Response**: Typo fixed.
*\* p10, L10: Reformulate this sentence*

**Response**: We have modified p10, L10 and now it reads:

> Moreover, Fig. 4 and 6 show that when crashes were substituted using the RBF technique, the STAR-VARS algorithm estimated the sensitivity indices of the most important parameters (FRAC, FC, C0) (Fig. 4) and less important parameters (LP, ETF, beta, K2) (Fig. 6) with high degrees of accuracy and robustness.

*\* p11, L1: Add "the" before "Four" and before "water"*
**Response**: Fixed.
*\* p11, L10: Replace "with" by "between these"*
**Response**: Corrected.
*\* p11, L12: Add "a" before "vegetation"*
**Response**: Fixed.
*\* p11, L13: Add "the" before "soil"*
**Response**: Fixed.
*\* p11, L21: Reformulate "As shown"*

**Response**: It has been edited as follows:

> The STAR-VARS algorithm identified these parameters as weakly influential (very low IVARS-50 values) using the proposed crash handling techniques. However, the associated sensitivity indices obtained by the RBF imputation method are about two orders of magnitude larger for the parameters in the left panel (Fig.13 (a, c)) and about four orders of magnitude larger for the parameters in the right panel (Fig. 13 (b, d)) compared to those obtained by the single NN and median substitution methods.

*\* p11, L23: Add an "s" to "order"*
**Response**: Fixed.
*\* p11, L29: Remove the "7"*
**Response**: Fixed.
*\* p13, L30: Replace "depends" by "depending"*
**Response**: Typo fixed.
*\* p14, L1: Which feature? Reformulate this sentence*
**Response**: Corrected.
*\* p14, L6: Replace "are" by "should be"*
**Response**: Corrected.
*\* p14, L7: Add an "s" to "problem"*
**Response**: Typo fixed.
*\* p14, L25: Remove the "s" from "involves"*
**Response**: Typo fixed.
*\* p14, L31: "The efficiency of our proposed simulation based strategies was shown: : :"*
**Response**: Fixed.

*\* p14, L30: This is a very long, complex sentence.*

**Response**: This sentence has been reformulated in the revised manuscript and now it reads:

The high efficiency of our proposed substitution-based approach is of prominent importance, particularly when dealing with GSA of the computationally expensive models mainly because our proposed approach does not need repeating the entire experiment.

*p15, L11: "causing" instead of "casing"*
**Response**: Typo fixed.
*p15, L18: "understanding" instead of "understand"*
**Response**: Typo fixed.
*p23 caption: Remove "C0" from the list of "moderately influential parameters"*
**Response**: Corrected.

*The authors argue to substitute data of failed simulation members in large ensemble simulations conducted for global parametric sensitivity analysis of dynamical earth system models. It is common for the models to crash for certain parameter value combinations that are randomly sampled from multidimensional parameter space using standard automated techniques. Using case studies, the authors show that it may be better to fill in the data from the failed experiments with data substitution techniques rather than the general practice of ignoring those experiments completely. The paper is generally well written and motivated. I point out my concerns below.*

We greatly appreciate Reviewer 2 for reviewing the manuscript and providing positive evaluations.

*1. The authors motivate the study well (Section 1.2). However, the authors state that the automated sampling method that they use - STAR-VARS breaks down if there are failed simulations for certain parameter combinations (Section 2.3). They do not provide a good reasoning for that, which I think is warranted. Are there other sampling methods that would not be sensitive to failed simulations? Why use STAR-VARS? Is the data substitution strategy only designed because of the limitation of STAR-VARS?*

Thanks for this comment. As we mentioned in **Section 1.2**, those GSA techniques that use a sampling strategy with a specific structure will fail if the simulation model crashes at certain parameter configurations such as the widely-used variance-based method of Saltelli et al. (2010). To further explain, we have revised **Section 1.2** as follows:

> 3. Ignoring the crashed runs in GSA may only be seen relevant when using purely random (and independent) samples (i.e., Monte Carlo method). In such cases, if the model crashes at a given parameter set, one may simply exclude that parameter set or generate another random parameter set (at the expense of increased computational cost) that results in a successful simulation.

> 4. Some efficient sampling techniques follow specific spatial arrangements; examples include the variance-based GSA proposed by Saltelli et al. (2010) or STAR-VARS of Razavi and Gupta (2016b). In GSA enabled with such structured sampling techniques, we cannot ignore crashed simulations because excluding sample points associated with simulation crashes will distort the structure of the sample set, causing inaccurate estimation of sensitivity indices. As a result, the user may have to re-do a part or the entire experiment depending on the GSA implementation.

*2. The impact of three data substitution techniques are compared. However, the first two methods are overly simplistic, and one can argue that they would yield poorer results a-priori - for example, the median is definitely not a good approximation for parameter combinations that are in the distribution tails, which may be more likely to crash. I do not see why the authors chose to present the results from those methods as one of their main results. It is fine to include them, but I think it would have been more useful to include results from different surrogate models, e.g. kriging, neural networks etc., which may be better as models for data substitution.*

We certainly agree with reviewer that the median substitution is a very simple (perhaps naive) approach compared to other methods such as RBF. Nevertheless, we have adopted these methods considering our main goal in this study, which was finding simple and effective strategies for handling simulation failures. To improve the explanation, we have added the following statement to **Section 2.2.1**:

> In sampling-based optimization, one may assign a very poor objective function value (e.g., a very large objective function in the minimization case) to a crashed solution, similar to the big M method for handling optimization constraints (Camm et al., 1990). Our first strategy in the GSA context adopts such an approach. However, since replacing crashes with a big value can magnify the effect of the crashed runs in GSA, instead we suggest choosing a measure of central tendency such as

mean or median to minimize the impact of the implausible parameter configurations on the GSA results. Perhaps replacing each simulation crash with some "central" value is the easiest and a computationally simple method for imputation. Depending on the distribution of the model response variables Y, the central value can be median or mean. If the distribution of the model responses is not highly skewed, imputing the crashes with the mean of the non-missing values may work. However, if the distribution exhibits skewness, then the median may be a better replacement because the mean is sensitive to the outliers.

Regarding the application of other surrogate models (e.g., kriging, etc.) as we mentioned in **Section 2.2.3** depending on the complexity and dimensionality of the response surface, other types of metamodels can be incorporated into the proposed framework. We did not intend to compare the performance of different metamodelling techniques in this study, so we only applied the well-known RBF technique. Of course, this could be a potential direction for future research. To address this comment, we have added the following sentence to the conclusion section of the revised manuscript:

> Finally, another possible future direction is to apply and test other types of emulation techniques such as kriging and support vector machine in handling model crashes.

*3. The authors appear to consider simulation failure as numerical artefacts. It could well be that parameter combinations are unphysical resulting in genuine crashes. Substituting data for these model crashes would result in unrealistic sensitivity. Likewise, unrealistic parameter combinations could also result in successful runs without crashes distorting the sensitivity analysis. It will be good if the authors could discuss this. The authors discuss this partly in section 5.1 for MESH model while exploring the reasons of simulation failure, but do not seem to relate it to their substitution strategy which is their main point.*

Thanks for this very good comment. The following paragraph has been added to **Section 5.1** of the revised manuscript to improve the discussion:

> We conclude this section by highlighting a point that should receive careful attention when applying the substitution-based methods in handling model crashes. In addition to the numerical artefacts in simulation models, some combinations of parameter values, which may not be physically justified, can also lead to simulation failures. As a result, there is risks that substituting data for these crashed runs contaminate the assessment of parameter importance. Preventing this type of risks requires knowledge about the reasonable parameter ranges in DESMs. This type of crashes can be significantly reduced by selecting plausible ranges of parameters based on physical knowledge or information of the problem (a process referred to as "parameter space refinement" (see e.g., Li et al., 2019; Williamson et al., 2013)). However, DESMs often consist of many interacting, uncertain parameters, and therefore very little may be known a priori about the implausible regions of the parameter space.

*4. The title reads as if something useful can be done with simulations that crashed. But, the strategy of the paper is to actually substitute the failed simulations. The authors should think about revising the title so that its not too misleading.*

We greatly appreciate the reviewer for this valuable suggestion. The title has been modified in the revised manuscript as:

[revised manuscript text omitted]